# CommonScenes: Generating Commonsense 3D Indoor Scenes with Scene Graph Diffusion

Guangyao Zhai[1,2*]    Evin Pınar Örnek[1,2*]    Shun-Cheng Wu[1]    Yan Di[1†]
Federico Tombari[1,3]    Nassir Navab[1,2]    Benjamin Busam[1,2]

{guangyao.zhai,evin.oernek,yan.di}@tum.de

[1]Technical University of Munich    [2]Munich Center for Machine Learning    [3]Google

https://sites.google.com/view/commonscenes

## Abstract

Controllable scene synthesis aims to create interactive environments for numerous industrial use cases. Scene graphs provide a highly suitable interface to facilitate these applications by abstracting the scene context in a compact manner. Existing methods, reliant on retrieval from extensive databases or pre-trained shape embeddings, often overlook scene-object and object-object relationships, leading to inconsistent results due to their limited generation capacity. To address this issue, we present *CommonScenes*, a fully generative model that converts scene graphs into corresponding controllable 3D scenes, which are semantically realistic and conform to commonsense. Our pipeline consists of two branches, one predicting the overall scene layout via a variational auto-encoder and the other generating compatible shapes via latent diffusion, capturing global scene-object and local inter-object relationships in the scene graph while preserving shape diversity. The generated scenes can be manipulated by editing the input scene graph and sampling the noise in the diffusion model. Due to the lack of a scene graph dataset offering high-quality object-level meshes with relations, we also construct *SG-FRONT*, enriching the off-the-shelf indoor dataset 3D-FRONT with additional scene graph labels. Extensive experiments are conducted on SG-FRONT, where *CommonScenes* shows clear advantages over other methods regarding generation consistency, quality, and diversity. Codes and the dataset are available on the website.

## 1 Introduction

**C**ontrollable **S**cene **S**ynthesis (**CSS**) refers to the process of generating or synthesizing scenes in a way that allows for specific entities of the scene to be controlled or manipulated. Existing methods operate on images [62] or 3D scenes [34] varying by controlling mechanisms from input scene graphs [24] or text prompts [5]. Along with the development of deep learning techniques, CSS demonstrates great potential in applications like the film and video game industry [7], augmented and virtual reality [65], and robotics [72, 68]. For these applications, scene graphs provide a powerful tool to abstract scene content, including scene context and object relationships.

This paper investigates scene graph-based CSS for generating coherent 3D scenes characterized by layouts and object shapes consistent with the input scene graph. To achieve this goal, recent methods propose two lines of solutions. The first line of works optimizes scene layouts [35] and retrieves objects [16] from a given database (see Figure 1 (a)). Such retrieval-based approaches are

---

*The first two authors contributed equally. †Corresponding author.

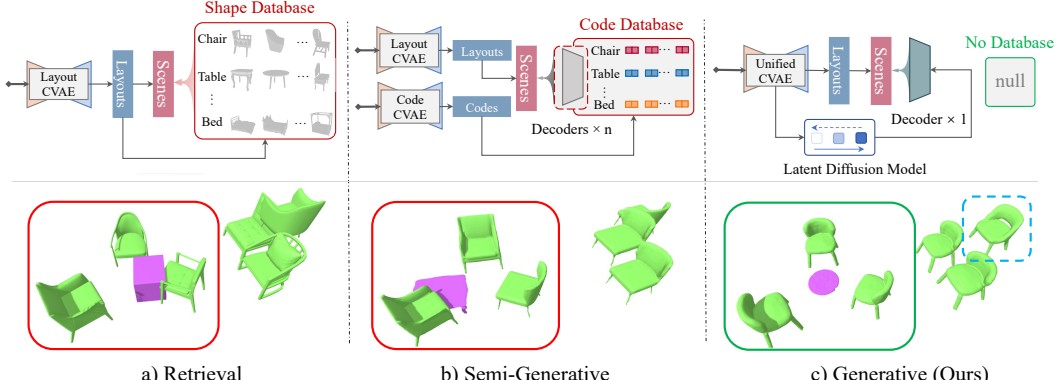

**Figure 1: Architecture Comparison (Upper Row):** Compared with previous methods, our fully generative model requires neither databases nor multiple category-level decoders. **Performance Comparison (Bottom Row):** We demonstrate the effectiveness of encapsulating scene-object and object-object relationships. The semantic information from the scene graph is *'a table is surrounded by three chairs'*. As highlighted in the rounded rectangles, through the scene-object relationship, our network outperforms other methods by generating a round table and three evenly distributed chairs. Through the object-object relationship, the three chairs are consistent in style. Moreover, our method still preserves the object diversity (blue dashed rectangle).

inherently sub-optimal [57] in generation quality due to performance limitations through the size of the database. The second solution, e.g., Graph-to-3D [12], regresses both the layout and shape of the objects for synthesis. However, the shape generation relies on pre-trained shape codes from category-wise auto-decoders, e.g., DeepSDF [42]. This semi-generative design (Figure 1 (b)) results in reduced shape diversity in the generated outputs. To enhance generation diversity without relying on vast databases, one possible solution is to concurrently predict scene layouts and generate object shapes with text-driven 3D diffusion models[8, 32], where the textual information is obtained from input scene graphs. Yet, in our experiments, we observe that such an intuitive algorithm works poorly since it does not exploit global and local relationship cues among objects encompassed by the graph.

In this work, our approach *CommonScenes* exploits global and local scene-object relationships and demonstrates that a fully generative approach can effectively encapsulate and generate plausible 3D scenes without prior databases. Given a scene graph, during training, we first enhance it with pre-trained visual-language model features, e.g., CLIP [46], and bounding box embeddings, incorporating coarse local inter-object relationships into the feature of each node. Then, we leverage a triplet-GCN [35] based framework to propagate information among objects, learning layouts through global cues and fine local inter-object relationships, which condition the diffusion process [47] to model the shape distribution as well. During inference, each node in the scene graph is enriched with the learned local-to-global context and sequentially fed into the latent diffusion model for each shape generation. Figure 1 (c) illustrates that our method effectively leverages relationship encoding to generate commonsense scenes, i.e., arranged plausibly and realistically, exhibiting scene-level consistency while preserving object shape diversity. Furthermore, to facilitate the benchmarking of CSS, we curate a novel indoor scene graph dataset, *SG-FRONT*, upon a synthetic dataset 3D-FRONT [18], since no existing indoor scene graph datasets provide high-quality meshes. SG-FRONT comprises around 45K 3D samples with annotated semantic and instance segmentation labels and a corresponding scene graph describing each scene.

Our contributions can be summarized into three points. **First**, we present *CommonScenes*, a fully generative model that converts scene graphs into corresponding 3D scenes using a diffusion model. It can be intuitively manipulated through graph editing. **Second**, *CommonScenes* concurrently models scene layout and shape distribution. It thereby encapsulates both global inter-object relationships and local shape cues. **Third**, we contribute *SG-FRONT*, a synthetic indoor dataset extending 3D-FRONT by scene graphs, thereby contributing graph-conditional scene generation benchmarks.

## 2 Related Work

**Scene Graph** Scene graphs provide a rich symbolic and semantic representation of the scene using nodes and relationships [25]. They are useful in many 2D-related topics such as image

generation [24, 67], image manipulation [12], caption generation [31], visual question answering [55], and camera localization [27]. Quickly after this progress, they are used in following areas: 3D scene understanding [59, 1, 64, 29], dynamic modeling [49], robotic grounding [22, 50, 68], spatio-temporal 4D [75, 30, 74], and controllable scene synthesis [33, 73, 60, 41, 13].

**Indoor 3D Scene Synthesis**  Controllable scene synthesis is extensively explored in the computer graphics community, varying from text-based scene generation [19] to segmentation map [43] or spatial layout-based image-to-image translation tasks [71]. Likewise in 3D, several methods generated scenes from images [56, 39, 2, 69, 14, 40, 38], text [36], probabilistic grammars [4, 23, 11, 45], layouts [26, 54], in an autoregressive manner [44, 63], or through learning deep priors [61]. Another line of work closer to ours is based on graph conditioning [33, 73, 60, 41]. Luo et al. [35] proposed a generative scene synthesis through variational modeling coupled with differentiable rendering. However, their method relies on shape retrieval, which depends on an existing database. Graph-to-3D [13] proposes a joint approach to learn both scene layout and shapes with a scene graph condition. Nevertheless, their object generation relies on a pre-trained shape decoder, limiting the generalizability. Unlike previous work, our method generates 3D scenes with the condition over a scene graph, which is trained end-to-end along with content awareness, resulting in higher variety and coherency.

**Denoising Diffusion Models**  A diffusion probabilistic model is a trained Markov chain that generates samples matching data by reversing a process that gradually adds noise until the signal vanishes [52]. Diffusion models have quickly gained popularity due to their unbounded, realistic, and flexible generation capacity [21, 53, 37, 28, 47, 15]. However, studies have identified that the diffusion models lack compositional understanding of the input text [51]. Several advancements have been introduced to address these limitations. Techniques such as the introduction of generalizable conditioning and instruction mechanisms have emerged [5, 70]. Moreover, optimizing attention channels during testing has also been explored [6, 17]. Recently, a latent diffusion model using a Signed Distance Field (SDF) to represent 3D shapes was proposed contemporaneously at multiple works [8, 32], which can be conditioned on a text or a single view image. For the contextual generation conditioned on scene graphs, methods have converted triplets into the text to condition the model [17, 67], where Yang et al. [67] proposed a graph conditional image generation based on masked contrastive graph training. To the best of our knowledge, we are the first to leverage both areas of scene graph and latent diffusion for end-to-end 3D scene generation.

## 3  Preliminaries

**Scene Graph**  A scene graph, represented as $\mathcal{G} = (\mathcal{V}, \mathcal{E})$, is a structured representation of a visual scene where $\mathcal{V} = \{v_i \mid i \in \{1, \ldots, N\}\}$ denotes the set of vertices (object nodes) and $\mathcal{E} = \{e_{i \to j} \mid i, j \in \{1, \ldots, N\}, i \neq j\}$ represents the set of directed edge from node $v_i$ to $v_j$. Each vertex $v_i$ is categorized through an object class $c_i^{node} \in \mathcal{C}^{node}$, where $\mathcal{C}^{node}$ denotes the set of object classes. The directed edges in $\mathcal{E}$ capture the relationships between objects in terms of both semantic and geometric information. Each edge $e_{i \to j}$ has a predicate class $c_{i \to j}^{edge} \in \mathcal{C}^{edge}$, where $\mathcal{C}^{edge}$ denotes the set of edge predicates. These relationships can incorporate various aspects, such as spatial locations (e.g., left/right, close by) or object properties (bigger/smaller). To facilitate subsequent processing and analysis, each node $v_i$ and edge $e_{i \to j}$ are typically transformed into learnable vectors $o_i$ and $\tau_{i \to j}$, respectively, through embedding layers, as shown in Figure 2.A.

**Conditional Latent Diffusion Model**  Diffusion models learn a target distribution by reversing a progressive noise diffusion process modeled by a fixed Markov Chain of length $T$ [21, 53]. Fundamentally, given a sample $\mathbf{x}_t$ from the latent space, gradual Gaussian Noise is added with a predefined scheduler $\mathbf{x}_t, t \in \{1, \ldots, T\}$ [21]. Then, the denoiser $\varepsilon_\theta$, typically a UNet[48], is trained to recover denoising from those samples. The recently introduced Latent Diffusion Models (LDMs) [47] reduce the computational requirements by learning this distribution in a latent space established by a pre-trained VQ-VAE [58] instead of directly processing the full-size input. The popular usage of LDM is conditional LDM, which allows the generation to obey the input cue $\mathbf{c}_i$ [47]. The training objective can be simplified to

$$\mathcal{L}_{LDM} = \mathbb{E}_{\mathbf{x}, \varepsilon \sim \mathcal{N}(0,1), t} \left[ ||\varepsilon - \varepsilon_\theta(\mathbf{x}_t, t, \mathbf{c}_i)||_2^2 \right], \tag{1}$$

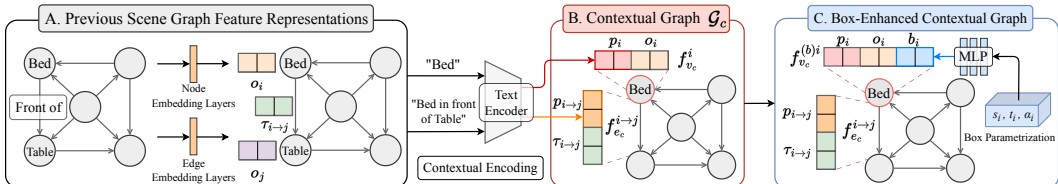

**Figure 2: Scene Graph Evolution.** Take the features of two nodes *Bed* ($o_i$), *Table* ($o_j$) and the linked edge *In front of* ($\tau_{i \to j}$) as an example, where $(o_i, o_j), \tau_{i \to j}$ are embedded learnable node features and the edge feature, respectively. We enhance the node and edge features with CLIP feature $p_i, p_{i \to j}$ to obtain *B. Contextual Graph*. Then, we parameterize the ground truth bounding box $b_i$ to the node to further build *C. Box-Enhanced Contextual Graph* with node and edge feature represented as $f_{v_c}^{(b)i} = \{p_i, o_i, b_i\}, f_{e_c}^{i \to j} = \{p_{i \to j}, \tau_{i \to j}\}$.

where $\mathbf{c}_i$ denotes a conditioning vector corresponding to the input sample $i$ fed into $\varepsilon_\theta$.

## 4 Method

**Overview** Given a semantic scene graph, our approach endeavors to generate corresponding 3D scenes conforming to commonsense. We employ a dual-branch network starting with a contextual encoder $E_c$, as shown in Figure 3. The two branches, referred to as the *Layout Branch* and the *Shape Branch*, function simultaneously for layout regression and shape generation. In Sec. 4.1, we first illustrate how a scene graph evolves to a **B**ox-enhanced **C**ontextual **G**raph (BCG) with features from pre-trained visual-language model CLIP [46] and bounding box parameters (Figure 2). Then, we show how BCG is encoded by $E_c$ and manipulated by the graph manipulator (Figure 3.A, B and C). In Sec. 4.2, we introduce the layout branch for layout decoding, and in Sec. 4.3, we introduce the shape branch for shape generation. Finally, we explain the joint optimization in Sec. 4.4.

### 4.1 Scene Graph Evolution

**Contextual Graph** As shown in Figure 2.A and B, we incorporate readily available prompt features from CLIP [46] as semantic anchors, capturing coarse inter-object information, into each node and edge of the input graph to conceptualize it as *Contextual Graph* $\mathcal{G}_c$ with $p_i = E_{\text{CLIP}}(c_i^{node})$ for objects and $p_{i \to j} = E_{\text{CLIP}}(c_i^{node} \boxplus c_{i \to j}^{edge} \boxplus c_j^{node})$ for edges. Here $E_{\text{CLIP}}$ is the pre-trained and frozen text encoder in [46], $\boxplus$ denotes the aggregation operation on prompts including subject class $c_i^{node}$, predicate $c_{i \to j}^{edge}$, and object class $c_j^{node}$. Thereby, the embeddings of $\mathcal{G}_c = (\mathcal{V}_c, \mathcal{E}_c)$ is formalized as,

$$\mathcal{F}_{\mathcal{V}_c} = \{f_{v_c}^i = (p_i, o_i) \mid i \in \{1, \dots, N\}\}, \quad \mathcal{F}_{\mathcal{E}_c} = \{f_{e_c}^{i \to j} = (p_{i \to j}, \tau_{i \to j}) \mid i, j \in \{1, \dots, N\}\}, \quad (2)$$

where $\mathcal{F}_{\mathcal{V}_c}$ represents the set of object features and $\mathcal{F}_{\mathcal{E}_c}$ the edge features.

**Box-Enhanced Contextual Graph** In training, we enrich each node in the contextual graph by using ground truth bounding boxes parameterized by box sizes $s$, locations $t$, and the angular rotations along the vertical axis $\alpha$, yielding a BCG, as shown in Figure 2.C. Thereby, the node embeddings of BCG are represented by $\mathcal{F}_{\mathcal{V}_c}^{(b)} = \{f_{v_c}^{(b)i} = (p_i, o_i, b_i) \mid i \in \{1, \dots, N\}\}$, where $b_i$ is obtained by encoding $(s_i, t_i, \alpha_i)$ with MLPs. Note that BCG is only used during training, i.e., bounding box information is not needed during inference.

**Graph Encoding** BCG is encoded by the subsequent triplet-GCN-based contextual encoder $E_c$, which together with the layout decoder $D_l$ in Sec. 4.2 construct a Conditional Variational Autoencoder (CVAE) [35]. The encoding procedure is shown in Figure 3.A, B and C. Given a BCG, during training, we input the embeddings $\mathcal{F}_{\mathcal{V}_c}^{(b)}$ and $\mathcal{F}_{\mathcal{E}_c}$ into $E_c$ to obtain an *Updated Contextual Graph* with node and edge features represented as $(\mathcal{F}_{\mathcal{V}_c}^{(z)}, \mathcal{F}_{\mathcal{E}_c})$, which is also the beginning point of the inference route. Each layer of $E_c$ consists of two sequential MLPs $\{g_1, g_2\}$, where $g_1$ performs message passing between connected nodes and updates the edge features, $g_2$ aggregates features from all connected neighbors of each node and update its features, as shown in the follows:

$$(\psi_{v_i}^{l_g}, \phi_{e_{i \to j}}^{l_g+1}, \psi_{v_j}^{l_g}) = g_1(\phi_{v_i}^{l_g}, \phi_{e_{i \to j}}^{l_g}, \phi_{v_j}^{l_g}), \quad l_g = 0, \dots, L-1,$$
$$\phi_{v_i}^{l_g+1} = \psi_{v_i}^{l_g} + g_2\Big(\text{AVG}\big(\psi_{v_j}^{l_g} \mid v_j \in N_{\mathcal{G}}(v_i)\big)\Big), \quad (3)$$

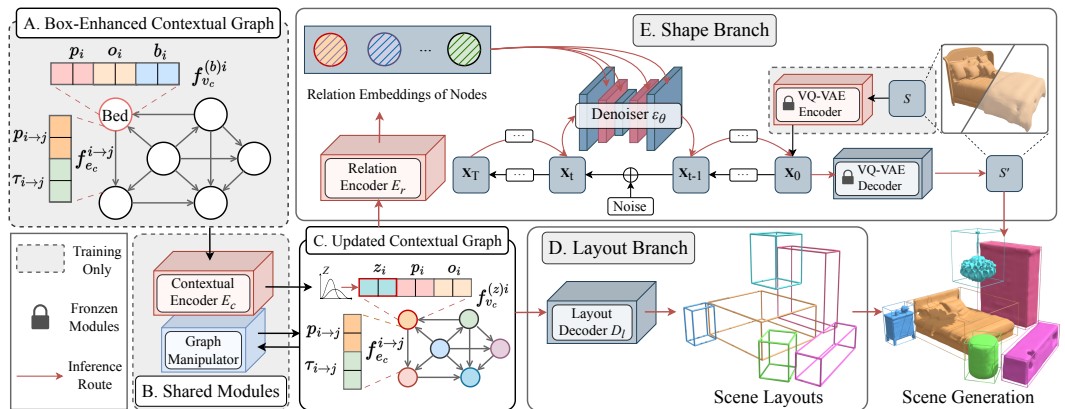

**Figure 3: Overview of CommonScenes.** Our pipeline consists of shared modules and two collaborative branches *Layout Branch* and *Shape Branch*. Given a BCG (Figure 2.C), we first feed it into $E_c$, yielding a joint layout-shape distribution $Z$. We sample $z_i$ from $Z$ for each node, obtaining concatenated feature $\{z_i, p_i, o_i\}$ with CLIP feature $p_i$ and self-updated feature $o_i$. A graph manipulator is then optionally adopted to manipulate the graph for data augmentation. Next, the updated contextual graph is fed into the layout branch and shape branch for layout regression and shape generation respectively. In the shape branch, we leverage $E_r$ to encapsulate global scene-object and local object-object relationships into graph nodes, which are then conditioned to $\varepsilon_\theta$ in LDM via cross-attention mechanism to generate $\mathbf{x}_0$ back in $T$ steps. Finally, a frozen shape decoder (VQ-VAE) reconstructs $S'$ using $\mathbf{x}_0$. The final scene is generated by fitting $S'$ to layouts.

where $l_g$ denotes a single layer in $E_c$ and $N_\mathcal{G}(v_i)$ includes all the connected neighbors of $v_i$. AVG refers to average pooling. We initialize the input embeddings of layer $0$ as the features from the *Updated Contextual Graph*, $(\phi_{v_i}^0, \phi_{e_{i \to j}}^0, \phi_{v_j}^0) = (f_{v_c}^{(b)i}, f_{e_c}^{i \to j}, f_{v_c}^{(b)j})$. The final embedding $\phi_{v_i}^L$ is used to model a joint layout-shape distribution $Z$, parameterized with the $N_c$-dimensional Gaussian distribution $Z \sim N(\mu, \sigma)$, where $\mu, \sigma \in \mathbb{R}^{N_c}$, predicted by two separated MLP heads. Thereby, $Z$ is modeled through minimizing:

$$\mathcal{L}_{KL} = D_{KL}\left(E_c^\theta(z|x, \mathcal{F}_{\mathcal{V}_c}^{(b)}, \mathcal{F}_{\mathcal{E}_c}) \,||\, p(z|x)\right), \tag{4}$$

where $D_{KL}$ is the Kullback-Liebler divergence measuring the discrepancy between $Z$ and the posterior distribution $p(z|x)$ chosen to be standard $N(z \,|\, 0, 1)$. We sample a random vector $z_i$ from $Z$ for each node, update node feature $\mathcal{F}_{\mathcal{V}_c}^{(z)}$ by concatenating $\{z_i, p_i, o_i\}$, keep edge features unchanged.

## 4.2 Layout Branch

As shown in Figure 3.D, a decoder $D_l$, which is another triplet-GCN, generates 3D layout predictions upon updated embeddings. In this branch, $E_c$ and $D_l$ are jointly optimized by Eq. (4) and a bounding box reconstruction loss:

$$\mathcal{L}_{layout} = \frac{1}{N} \sum_{i=1}^{N} (|s_i - \hat{s}_i|_1 + |t_i - \hat{t}_i|_1 - \sum_{\lambda=1}^{\Lambda} \alpha_i^\lambda \log \hat{\alpha}_i^\lambda), \tag{5}$$

where $\hat{s}_i, \hat{t}_i, \hat{\alpha}_i$ denote the predictions of bounding box size $s$, locations $t$ and angular rotations $\alpha$, respectively. $\lambda$ is the rotation classification label. We partition the rotation space into $\Lambda$ bins, transforming the rotation regression problem into a classification problem.

## 4.3 Shape Branch

As shown in Figure 3.E, in parallel to the *Layout Branch*, we introduce *Shape Branch* to generate shapes for each node in the given graph, and represent them by SDF.

**Relation Encoding** The core idea of our method is to exploit global scene-object and local object-object relationships to guide the generation process. Hence, besides incorporating CLIP features for coarse local object-object relationships, as introduced in Section 4.1, we further design a relation encoder $E_r$, based on triplet-GCN as well, to encapsulate global semantic cues of the graph into

each node and propagate local shape cues between connected nodes. Specifically, $E_r$ operates on the learned embeddings $(\mathcal{F}_{\mathcal{V}_c}^{(z)}, \mathcal{F}_{\mathcal{E}_c})$ of the *Updated Contextual Graph*, and updates the feature of each node with local-to-global semantic cues, yielding node-relation embeddings for subsequent diffusion-based shape generation.

**Shape Decoding**  We use an LDM conditioned on node-relation embedding to model the shape-generation process. In terms of shape representation, we opt for truncated SDF (TSDF) [10] around the surface of the target object within a voxelized space $S \in \mathbb{R}^{D \times D \times D}$. Following the LDM, the dimension $D$ of TSDF can be reduced by training a VQ-VAE [58] as a shape compressor to encode the 3D shape into latent dimensions $\mathbf{x} \in \mathbb{R}^{d \times d \times d}$, where $d << D$ is built upon a discretized codebook. Then a forward diffusion process adds random noise to the input shape $\mathbf{x}_0$ transferring to $\mathbf{x}_T$, upon which we deploy a 3D-UNet [9] $\varepsilon_\theta$ (Figure 3.E) to denoise the latent code back to $\mathbf{x}_0$ according to DDPM model by Ho et al. [21]. The denoiser is conditioned on node-relation embeddings to intermediate features of 3D UNet via the cross-attention mechanism. Finally, the decoder of VQ-VAE generates the shape $S'$ from the reconstructed $\mathbf{x}_0$. For the denoising process at the timestep $t$, the training objective is to minimize:

$$\mathcal{L}_{shape} = \mathbb{E}_{\mathbf{x}, \varepsilon \sim \mathcal{N}(0,1), t} \left[ ||\varepsilon - \varepsilon_\theta(\mathbf{x}_t, t, E_r(\mathcal{F}_{\mathcal{V}_c}^{(z)}, \mathcal{F}_{\mathcal{E}_c})||_2^2 \right], \tag{6}$$

where the evidence lower bound between the sampled noise $\varepsilon$ and the prediction conditioned on the contextual relation embedding extracted from $E_r$ is optimized. At test time, a latent vector is randomly sampled from $\mathcal{N}(0,1)$ and progressively denoised by 3D-Unet to generate the final shape. Each shape within the layout is populated based on per-node conditioning, ultimately producing a plausible scene. Compared to prior work, our design can bring more diverse shape generation by taking advantage of the diffusion model architecture, as shown in experiments.

### 4.4 Layout-Shape Training

Our pipeline is trained jointly in an end-to-end fashion, allowing the *Layout Branch* and *Shape Branch* to optimize each other by sharing latent embeddings $(\mathcal{F}_{\mathcal{V}_c}^{(z)}, \mathcal{F}_{\mathcal{E}_c})$, coming out of $E_c$. The final optimization loss is the combination of scene distribution modeling, layout generation, and shape generation:

$$\mathcal{L} = \lambda_1 \mathcal{L}_{KL} + \lambda_2 \mathcal{L}_{layout} + \lambda_3 \mathcal{L}_{shape}, \tag{7}$$

where $\lambda_1$, $\lambda_2$ and $\lambda_3$ are weighting factors. Further insights on accomplishing the batched training are provided in Supplementary Material.

## 5  Experiments

**SG-FRONT Dataset**  Due to the lack of scene graph datasets also providing high-quality object meshes, we construct *SG-FRONT*, a set of well-annotated scene graph labels, based on a 3D synthetic dataset 3D-FRONT [18] that offers professionally decorated household scenarios. The annotation labels can be grouped into three categories ***spatial/proximity, support, and style***. The spatial relationships are based on a series of relationship checks between the objects, which control the object bounding box location, e.g., `left`/`right`, the comparative volumes, e.g., `bigger`/`smaller`, and heights, e.g., `taller`/`shorter`. The support relationships include directed structural support, e.g., `close by`, `above`, and `standing on`. Thresholds for these labels are iterated and set by the annotators. Finally, style relationships include the object attributes to assign labels, namely `same material as`, `same shape as`, and `same super-category as`. SG-FRONT contains around 45K samples from three different indoor scene types, covering 15 relationship types densely annotating scenes. More details are provided in the Supplementary Material.

**Implementation Details**  We conduct the training, evaluation, and visualization of *CommonScenes* on a single NVIDIA A100 GPU with 40GB memory. We adopt the AdamW optimizer with an initial learning rate of 1e-4 to train the network in an end-to-end manner. We set $\{\lambda_1, \lambda_2, \lambda_3\} = \{1.0, 1.0, 1.0\}$ in all our experiments. $N_c$ in distribution $Z$ is set to 128 and TSDF size $D$ is set as 64. We provide more details in the Supplementary Material.

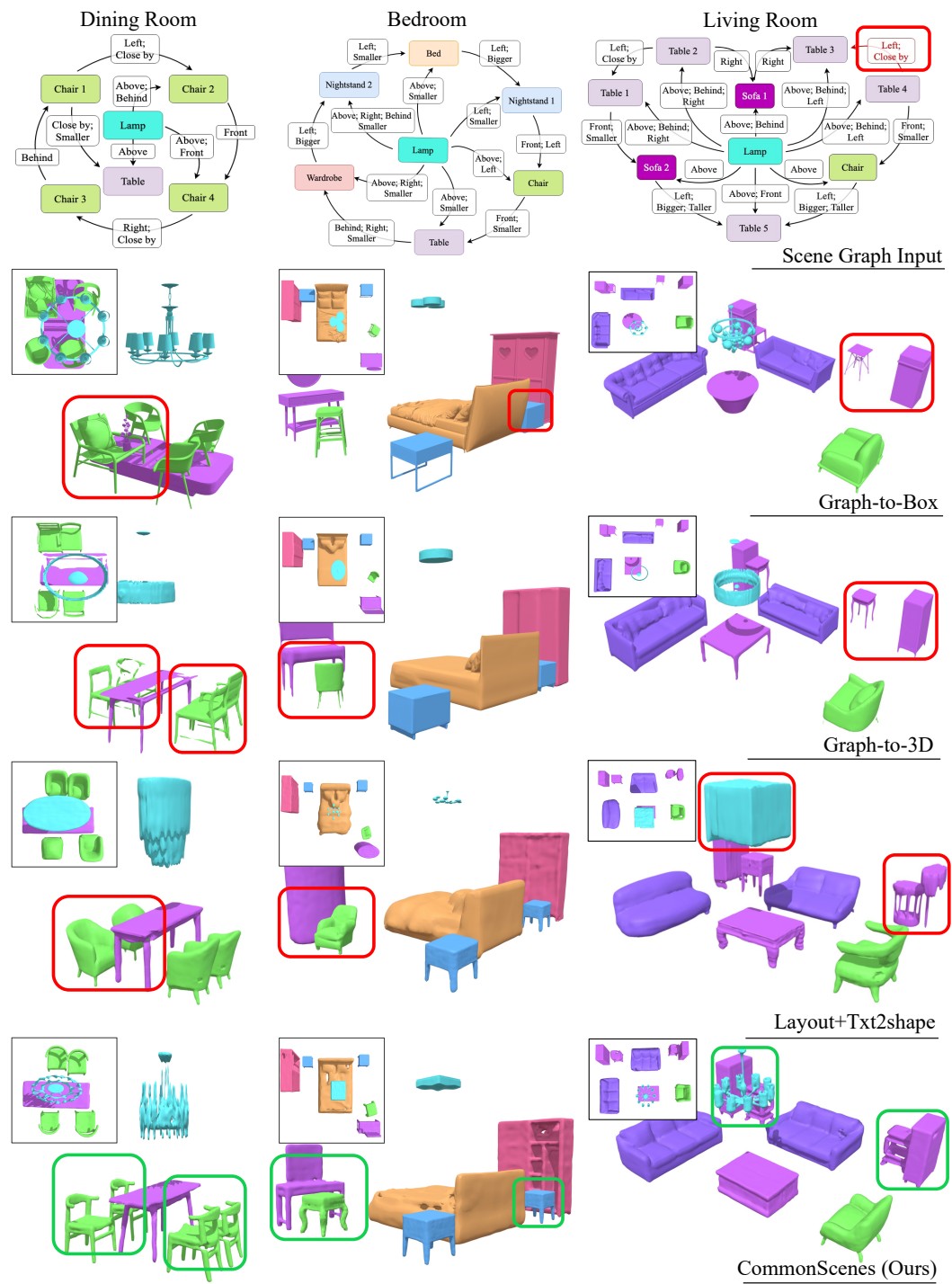

**Figure 4: Qualitative comparison** The orientations of Left/Right and Front/Behind in the scene graph align with the top-down view. Both scene-object and object-object inconsistencies are highlighted in red rectangles. Green rectangles emphasize the commonsense consistency our method produces.

**Evaluation Metrics** To measure the **fidelity and diversity** of generated scenes, we employ the commonly adopted Fréchet Inception Distance (FID) [20] & Kernel Inception Distance (KID) [3] metrics [44]. We project scenes onto a bird's-eye view, excluding lamps to prevent occlusion and using object colors as semantic indicators. To measure the **scene graph consistency**, we follow the scene graph constraints [13], which measure the accuracy of a set

| Method | Shape Representation | Bedroom FID | Bedroom KID | Living room FID | Living room KID | Dining room FID | Dining room KID | All FID | All KID |
|---|---|---|---|---|---|---|---|---|---|
| 3D-SLN [35] | | 57.90 | 3.85 | 77.82 | 3.65 | 69.13 | 6.23 | 44.77 | 3.32 |
| Progressive [13] | Retrieval | 58.01 | 7.36 | 79.84 | 4.24 | 71.35 | 6.21 | 46.36 | 4.57 |
| Graph-to-Box [13] | Retrieval | 54.61 | 2.93 | 78.53 | 3.32 | 67.80 | 6.30 | 43.51 | 3.07 |
| **Ours** w/o SB | | **52.69** | **2.82** | **76.52** | **2.08** | **65.10** | **6.11** | **42.07** | **2.23** |
| Graph-to-3D [13] | DeepSDF [42] | 63.72 | 17.02 | 82.96 | 11.07 | 72.51 | 12.74 | 50.29 | 7.96 |
| Layout+txt2shape | SDFusion [8] | 68.08 | 18.64 | 85.38 | 10.04 | **64.02** | **5.08** | 50.58 | 8.33 |
| **Ours** | rel2shape | **57.68** | **6.59** | **80.99** | **6.39** | 65.71 | 5.47 | **45.70** | **3.84** |

**Table 1: Scene generation realism** as measured by FID and KID ($\times 0.001$) scores at $256^2$ pixels between the top-down rendering of generated and real scenes (lower is better). Two main rows are separated with respect to the reliance on an external shape database for retrieval. "Ours w/o SB" refers to ours without the shape branch.

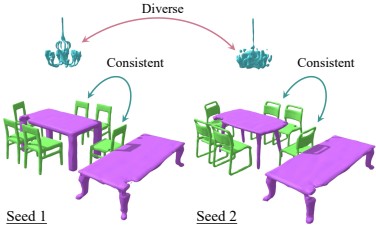

**Figure 5: Consistency co-exists with diversity in different rounds.** Our generated objects show diversity when activated twice while preserving the shape consistency within the scene (chairs in a suit).

| Method | Consistency ↓ Chair | Consistency ↓ Table | Diversity ↑ Chair | Diversity ↑ Table |
|---|---|---|---|---|
| Graph-to-Box [13] | 10.42 | 50.66 | 0.53 | 0.89 |
| Graph-to-3D [13] | 2.49 | 11.74 | 1.43 | 0.93 |
| Ours | **1.96** | **9.04** | **30.04** | **10.53** |

**Table 2: Consistency and diversity in the dining rooms.** The object shapes related with *same as* within a scene are consistent as indicated by low CD values ($\times 0.001$), whereas the shapes across different runs have high diversity, per high KL divergence.

of relations on a generated layout. We calculate the spatial `left/right`, `front/behind`, `smaller/larger`, `taller/shorter` metrics, as well as more difficult proximity constraints `close by` and `symmetrical`. To measure the **shape consistency**, we test dining rooms, a typical scenario in which tables and chairs should be in suits according to the commonsense decoration. We identify matching dining chairs and tables in 3D-FRONT using the same CAD models and note their instance IDs. This helps us determine which entities belong to the same sets. We calculate Chamfer Distance (CD) of each of the two objects in each set after generation. To measure the **shape diversity**, we follow [13] to generate each scene 10 times and evaluate the change of corresponding shapes using CD. We illustrate the concepts of diversity and consistency in Figure 5. We also report MMD, COV, and 1-NNA following [66] for **the object-level evaluation** in the Supplementary Materials.

**Compared Baselines**  We include three types of baseline methods. **First**, three retrieval-based methods, i.e., a layout generation network *3D-SLN* [35], a progressive method to add objects one-by-one designed in [13], and *Graph-to-Box* from [13]. **Second**, a semi-generative SOTA method *Graph-to-3D*. **Third**, an intuitive method called *layout+txt2shape* that we design according to the instruction in Sec. 1, stacking a layout network and a text-to-shape generation model in series, and it is a fully generative approach with text only. All baseline methods are trained on the SG-FRONT dataset following the public implementation and training details.

## 5.1 Graph Conditioned Scene Generation

**Qualitative results**  The generated scenes from different baselines are shown in Figure 4: (a) Graph-to-Box retrieval, (b) Graph-to-3D, (c) layout+txt2shape, and (d) ours. It can be seen that our method has better shape consistency and diversity. Specifically, in terms of object shape quality, retrieval baseline scenes look optimal since they are based on real CAD models. Yet, the layout configuration is poor, and the retrieved object styles are not plausible, e.g., dining chairs and nightstands are not in a suit, respectively, in the same rooms. Graph-to-3D improves this by learning coherent scenes. However, the object styles are not conserved within scene styles, providing unrealistic scene-object inconsistency, e.g., in the bedroom, the height of the chair does not match the style and the height of the table. Subsequently, (c) improves the layout, which can be seen in the living room area, but again falls back on the generated shape quality and shares the same problem with Graph-to-Box on the poor object-object consistency. In contrast, CommonScenes can capture diverse environments considering both generation consistency and stylistic differences with respect to each scene. In the

| Method | Shape Representation | Easy | | | | Hard* | |
|---|---|---|---|---|---|---|---|
| | | left / right | front / behind | smaller / larger | taller / shorter | **close by**∗ | **symmetrical**∗ |
| 3D-SLN [35] | | 0.97 | 0.99 | 0.95 | 0.91 | 0.72 | 0.47 |
| Progressive [13] | Retrieval | 0.97 | 0.99 | 0.95 | 0.82 | 0.69 | 0.46 |
| Graph-to-Box [13] | | **0.98** | 0.99 | 0.96 | **0.95** | 0.72 | 0.45 |
| **Ours** w/o SB | | **0.98** | 0.99 | **0.97** | **0.95** | 0.74 | **0.63** |
| Graph-to-3D [13] | DeepSDF [42] | **0.98** | 0.99 | **0.97** | **0.95** | 0.74 | 0.57 |
| **Ours** | rel2shape | **0.98** | **1.00** | **0.97** | **0.95** | **0.77** | 0.60 |

**Table 3:** Scene graph constrains on the **generation** task (higher is better). The total accuracy is computed as the mean over the individual edge class accuracy to minimize class imbalance bias.

| Method | Shape Representation | Mode | Easy | | | | Hard* | |
|---|---|---|---|---|---|---|---|---|
| | | | left /right | front / behind | smaller / larger | taller / shorter | **close by**∗ | **symmetrical**∗ |
| 3D-SLN [35] | | | 0.89 | 0.90 | 0.55 | 0.58 | 0.10 | 0.09 |
| Progressive [13] | Retrieval | | 0.89 | 0.89 | 0.52 | 0.55 | 0.08 | 0.09 |
| Graph-to-Box [13] | | | **0.91** | 0.91 | **0.86** | 0.91 | 0.66 | 0.53 |
| **Ours** w/o SB | | change | **0.91** | **0.92** | **0.86** | **0.92** | **0.70** | 0.53 |
| Graph-to-3D [13] | DeepSDF [42] | | **0.91** | **0.92** | **0.86** | 0.89 | 0.69 | 0.46 |
| **Ours** | rel2shape | | **0.91** | **0.92** | **0.86** | 0.91 | 0.69 | **0.59** |
| 3D-SLN [35] | | | 0.92 | 0.92 | 0.56 | 0.58 | 0.05 | 0.05 |
| Progressive | Retrieval | | 0.92 | 0.91 | 0.53 | 0.54 | 0.02 | 0.06 |
| Graph-to-Box [13] | | | 0.94 | 0.93 | 0.90 | 0.94 | 0.67 | 0.58 |
| **Ours** w/o SB | | addition | **0.95** | **0.95** | 0.90 | 0.94 | **0.73** | **0.63** |
| Graph-to-3D [13] | DeepSDF [42] | | 0.94 | **0.95** | **0.91** | 0.93 | 0.63 | 0.47 |
| **Ours** | rel2shape | | **0.95** | **0.95** | **0.91** | **0.95** | 0.70 | 0.61 |

**Table 4:** Scene graph constraints on the **manipulation** task (higher is better). The total accuracy is computed as the mean over the individual edge class accuracy to minimize class imbalance bias. Top: Relationship change mode. Bottom: Node addition mode.

living room, the scene graph input requires that Table No.3 and Table No.4 should stand close to each other. Graph-to-Box and Graph-to-3D fail to achieve the goal, which is also reflected in Table 3 and Table 4 that these kinds of methods cannot handle `close by` relation. We show more results in Supplementary Material.

**Quantitative results** We provide the FID/KID scores in Table 1 separated as average over three room types. Our method establishes the best results among the free-shape representation methods by improving the Graph-to-3D by approximately 10% on FID and 40% on KID on average. On the other hand, most of the *layout+txt2shape* results are worse than the previous state-of-the-art methods, proving the requirement of learning layout and shape jointly. Notably, the retrieval-based methods overall have better scores than those of free shapes since the retrieved objects align well with the test database. Further on, among the retrieval baselines, ours without the diffusion shape branch (*Ours w/o SB*) surpasses the prior works, indicating the benefit of contextual graph representation. Additionally, on the generation consistency, we can observe in Table 2 that our method can reflect the edge relationship in the generated shapes directly, indicated by significant improvement in the consistency score. By leveraging the latent diffusion model, our diversity is significantly improved compared with other methods. We provide more results on diversity and also show perceptual study on the evaluation of consistency and diversity in the Supplementary Material.

In terms of scene graph constraints, we present the results as easy and hard metrics in Table 3. On easy metrics, our methods (Ours, Ours w/o SB) are either better than or on par with other methods. While on hard metrics, they are superior to others. It shows that the spatial relationships could be learned easier than the proximity (`close by`) and commonsense (`symmetrical`) based ones. The improvement over the `symmetrical` constraint is particularly evident, as our method surpasses the state-of-the-art (SOTA) by 10%. This demonstrates that the integrated layout and shape cues are crucial for learning these contextual configurations.

## 5.2 Scene Manipulation

We demonstrate the downstream application of scene manipulation on our method and compare it with the aforementioned methods on scene graph constraints in Table 4. Our methods have the highest total score in both relation change and object addition modes. In both modes, the results are

on par with Graph-to-3D on easy relations, whereas there is a major improvement in the hard ones. Further, within the retrieval methods, Ours w/o SB improves over the baselines in the addition mode and is on par in the change mode. It could be argued that changing could be more difficult since it could alter multiple objects, whereas object insertion has less effect on the overall scene. We show some qualitative manipulation examples in the Supplementary Material.

### 5.3 Ablations

We ablate the primary components of CommonScenes in Table 5, including (1) scene generation from the original scene graph without contextual information brought by CLIP features, (2) scene generation from the contextual graph without the participation of the relational encoder $E_r$, (3) conditioning with concatenation on diffusion latent code instead of cross-attention, and (4) our final proposed method. We provide the mean FID scores over the scenes, as well as the mean over the hard scene graph constraints (mSG). We observe the benefit of both the context and $E_r$ indicated by FID/KID scores, as well as the choice of our conditioning mechanism.

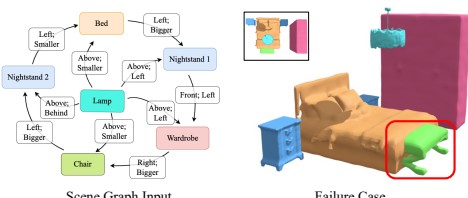

| Condition | FID | KID | mSG |
|---|---|---|---|
| Ours w/o context | 50.24 | 5.63 | 0.59 |
| Ours w/o $E_r$ | 51.62 | 5.41 | 0.73 |
| Ours with concat | 48.57 | 4.25 | 0.71 |
| **Ours** | **45.70** | **3.84** | **0.74** |

**Figure 6:** An interpenetrating phenomenon.       **Table 5:** Ablations under three circumstances.

## 6 Limitations

We address the main aspects of the dataset and the limitations of our method and discuss more in the Supplementary Materials. First, the 3D-FRONT dataset used in our research contains significant noise, which we have mitigated through a post-processing step. Despite this effort, a small proportion of noisy data, specifically interpenetrating furniture instances, remains in the training dataset. Consequently, our proposed method and the baseline approaches reflect this during inference, rarely resulting in occurrences of scenes with collided objects. We show some cases in Figure 6. While our method outperforms others by effectively combining shape and layout information, it is essential to note that minor collision issues may still arise.

## 7 Conclusion

Scene graph-based CSS designs interactive environments suitable for a wide range of usages. Current methods heavily depend on object retrieval or pre-trained shape models, neglecting inter-object relationships and resulting in inconsistent synthesis. To tackle this problem, we introduce *CommonScenes*, a fully generative model that transforms scene graphs into corresponding commonsense 3D scenes. Our model regresses scene layouts via a variational auto-encoder, while generating satisfying shapes through latent diffusion, gaining higher shape diversity and capturing the global scene-object relationships and local object-object relationships. Additionally, we annotate a scene graph dataset, *SG-FRONT*, providing object relationships compatible with high-quality object-level meshes. Extensive experiments on *SG-FRONT* show that our method outperforms other methods in terms of generation consistency, quality, and diversity.

**Acknowledgements**   This research is supported by Google unrestricted gift, the China Scholarship Council (CSC), and the Munich Center for Machine Learning (MCML). We are grateful to Google University Relationship GCP Credit Program for supporting this work by providing computational resources. Further, we thank all participants of the perceptual study.

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
