# Supplementary Material
# CommonScenes: Generating Commonsense 3D Indoor Scenes with Scene Graph Diffusion

**Guangyao Zhai**    **Evin Pınar Örnek**    **Shun-Cheng Wu**    **Yan Di**
**Federico Tombari**    **Nassir Navab**    **Benjamin Busam**

{guangyao.zhai,evin.oernek,yan.di}@tum.de

In this supplementary, we additionally report the following:

- Section 1: Additional results.
- Section 2: User perceptual studies.
- Section 3: SG-FRONT dataset details.
- Section 4: Results on 3DSSG dataset.
- Section 5: More qualitatives on scene generation.
- Section 6: Discussion and limitations.
- Section 7: Additional training details.

We further provide a supplementary video attached to this manuscript to provide spatio-temporal illustrations and further explanations of our method.

## 1 Additional Results

### 1.1 Diversity results.

In Table 1, we report the results of 10 categories tested on randomly selected 200 test scenes against the state-of-the-art semi-generative method Graph-to-3D [3] and its corresponding object retrieval method Graph-to-Box. We use Chamfer Distance ($\times 0.001$) as the metric, following the protocol in Graph-to-3D [3]. Specifically, we generate each scene 10 times and calculate an average of the chamfer distance between corresponding objects in adjacent scenes. Our method shows a significant improvement upon the diversity leveraging the merits of the diffusion model, with a qualitative example shown in Figure 1. The other two methods behave worse because they heavily rely on the shape and embedding of databases, which limits the generative ability.

| Method | Bed | Nightstand | Wardrobe | Chair | Table | Cabinet | Lamp | Shelf | Sofa | TV stand | Total |
|---|---|---|---|---|---|---|---|---|---|---|---|
| Graph-to-Box [3] | 0.66 | 0.01 | 0.18 | 0.33 | 1.30 | 1.16 | 1.57 | 0.32 | 0.71 | 0.01 | 0.53 |
| Graph-to-3D [3] | 1.01 | 2.06 | 0.87 | 1.61 | 0.83 | 0.96 | 0.70 | 1.25 | 0.97 | 1.42 | 1.21 |
| **Ours** | **39.59** | **68.78** | **20.01** | **46.03** | **112.53** | **32.28** | **140.56** | **191.55** | **58.58** | **58.08** | **73.40** |

Table 1: **Diversity performance** on SG-FRONT and 3D-FRONT.

### 1.2 Object-level evaluation.

Since our objective is scene generation, we use FID/KID as the main metrics for evaluating the scene-level generation quality in the main paper. Although object generation is not the main focus of

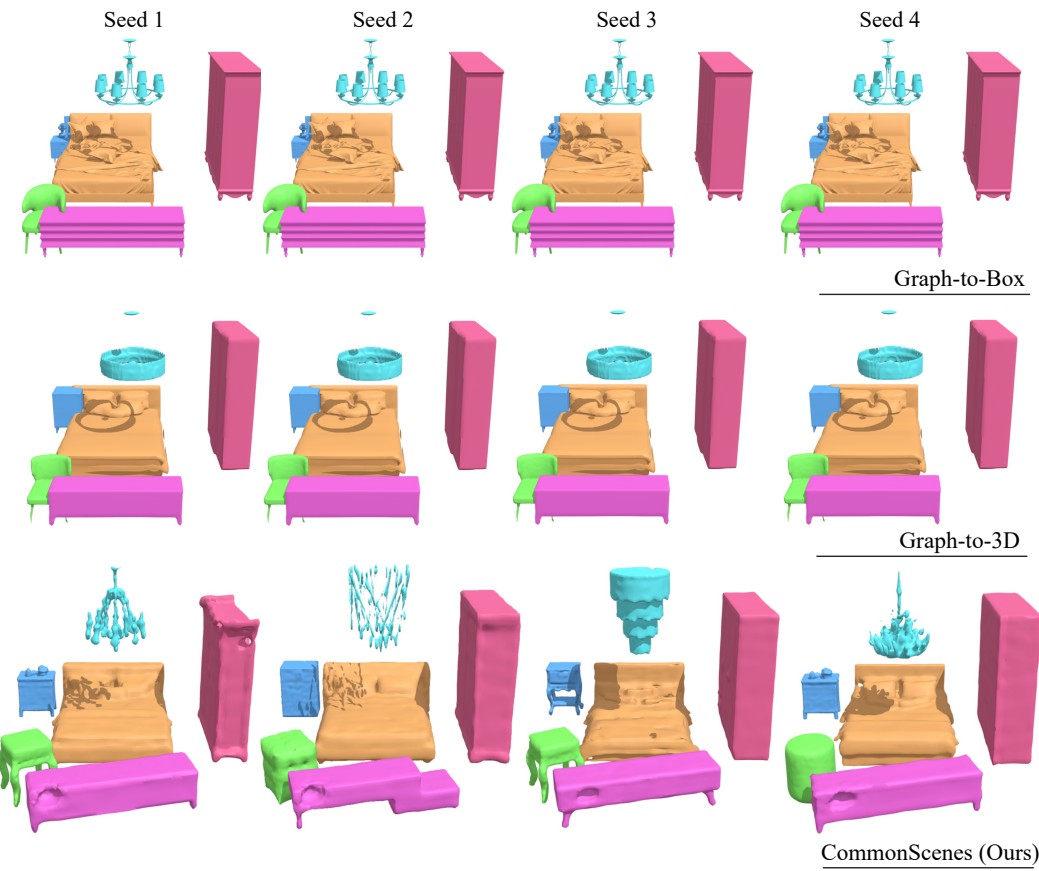

Figure 1: **Diversity comparison**. Our method shows a huge diversity when generating the same bedroom scene in different rounds.

this work, we believe the object-level analysis is valuable, since this is an integral component of the proposed model. We follow PointFlow [11] to report the MMD (×0.01) and COV (%) for evaluating per-object generation. We collect ground truth objects in each category within the test set. As shown in the first two rows of Table 2, our method shows better performance in both MMD and COV, which highlights the object-level shape generation ability of CommonScenes. We also calculate 1-nearest

| Method | Metric | Bed | Nightstand | Wardrobe | Chair | Table | Cabinet | Lamp | Shelf | Sofa | TV stand |
|---|---|---|---|---|---|---|---|---|---|---|---|
| Graph-to-3D [3] | MMD (↓) | 1.56 | 3.91 | 1.66 | 2.68 | 5.77 | 3.67 | 6.53 | 6.66 | 1.30 | 1.08 |
| **Ours** | | **0.49** | **0.92** | **0.54** | **0.99** | **1.91** | **0.96** | **1.50** | **2.73** | **0.57** | **0.29** |
| Graph-to-3D [3] | COV (%, ↑) | 4.32 | 1.42 | 5.04 | 6.90 | 6.03 | 3.45 | 2.59 | 13.33 | 0.86 | 1.86 |
| **Ours** | | **24.07** | **24.17** | **26.62** | **26.72** | **40.52** | **28.45** | **36.21** | **40.00** | **28.45** | **33.62** |
| Graph-to-3D [3] | 1-NNA (%, ↓) | 98.15 | 99.76 | 98.20 | 97.84 | 98.28 | 98.71 | 99.14 | 93.33 | 99.14 | 99.57 |
| **Ours** | | **85.49** | **95.26** | **88.13** | **86.21** | **75.00** | **80.17** | **71.55** | **66.67** | **85.34** | **78.88** |

Table 2: **Object-level generation performance.** We report MMD(×0.01), COV and 1-NNA for evaluating shapes by means of quality and diversity.

neighbor accuracy (1-NNA, %), which directly measures distributional similarity on both diversity and quality. The closer the 1-NNA is to 50%, the better the shape distribution is captured. It can be observed that our method surpasses Graph-to-3D in the evaluation of distributional similarity. Coupled with the results in Table 2 and Table 1, CommonScenes exhibits more plausible object-level generation than the previous state-of-the-art.

### 1.3 Manipulation results.

Our method inherits the manipulation function from Graph-to-3D, allowing users to manipulate the updated contextual graph during the training and inference phases. As shown in Figure 2, we provide samples of all three room types ('Bedroom', 'Living Room', 'Dining Room') for object addition and relation change modes. For instance, in the first scene of Figure 2. a) we first let the model generate six chairs, two tables, and a lamp. Then in the second round, we insert a chair and its edges into other objects in the scene, where the model can still generate the corresponding scenes. In the first scene of Figure 2. b) the model first generates two tables on the right and one cabinet on the left. In the second round, we manipulate the cabinet to the right side of the left table in the scene. It is an expected sign when the appearances of some objects change, as the second round enjoys different random layout-shape vectors and noise. Notably, the inserted objects preserve the stylistic consistency within the scene since the sampling is based on the existing contextual knowledge of the scene (e.g., sofa and chair insertions). This is also observed in relation to the change mode, where the generated objects are still realistic after the size or location change. For example, in the bedroom scene where the "nightstand, shorter, bed" triplet is changed to "nightstand, taller, bed", the method can insert plausible objects by replacing the bed with a shorter bed and the nightstand with a taller one, instead of simply stretching and deforming the existing shapes.

### 1.4 Statistical significance test

We re-run the inference process multiple times with random seeds to test the fluctuation in our scene generation results. Overall, after running the method over ten times, the mean FID and KID scores are in the same range as the results we reported in the main paper Table 1, and the variance in terms of standard deviation on FID is in average 0.05 scores, where the numbers for bedroom, dining room, and living room are 0.034, 0.021, and 0.07 respectively. This does not alter the comparative results provided in the paper.

## 2 User Perceptual Studies

We conducted a perceptual study to evaluate the quality of our generated scenes. We generated scenes from Graph-to-3D, and our method for this. At every step, we provide samples from paired methods and ask users to select the better one according to the criteria, (1) Global correctness and realism (*Does the arrangement of objects look correct?*), (2) functional and style fitness (*Does the scene look stylistically coherent?*), (3) scene graph correctness. Additionally, we asked for the number of errors in interpenetrating furniture for each scene to evaluate the visible errors. Three answers were shown (1) No errors, (2) One error, and (3) Multiple errors to prevent attention degradation of users. We randomly sample $\sim 20$ scenes covering all scene types. For each scene, we provide a top-down view rendering and a side-view rendering to ensure the visibility of the entire scene. We illustrate the user interface of this study in Figure 3.

Overall, $\sim 25$ subjects from various nationalities, professional backgrounds, and ages participated in our study. We provide the outcomes of this study in Figure 3 and Figure 4. The scenes generated from our method were preferred over the other methods $83\%$ for correctness and realism, $80\%$ for style fitness, and $87\%$ for correctness. Where layout+txt2shape was instead preferred $16\%$, $18\%$ and $6\%$ and Graph-to-3D $17\%$, $20\%$ and $15\%$, respectively. Interestingly, our method was preferred over the scene graph correctness much higher than other aspects, whereas layout+txt2shape was the lowest, proving the necessity of information sharing between layout and 3D shapes. Unsurprisingly, Graph-to-3D was preferred more than layout+txt2shape because of the same reason. The inclination towards our method was the highest in all aspects, indicating that our proposed full generative method could generate commonsense scenes.

Additionally, we can observe the error comparison in Figure 4 that the users indicated $84\%$ scenes having no error, $11\%$ one error, and $6\%$ as more than one error, where all others had $36\%$ none, $25\%$ one, and $39\%$ multiple errors. We also provide the statistics separately for each method. Here, it can be seen that on the scenes compared with the layout+txt2shape method, ours had $90\%$ no errors and $2\%$ multiple errors, whereas the compared method had $64\%$ none and $22\%$ multiple. In the scenes that were compared across Graph-to-3D and ours, they had $20\%$ no errors and $49\%$ multiple errors, whereas our method $80\%$ none and only $8\%$ multiple errors. This result shows that our method improves the error of interpenetrating objects compared to the other baselines.

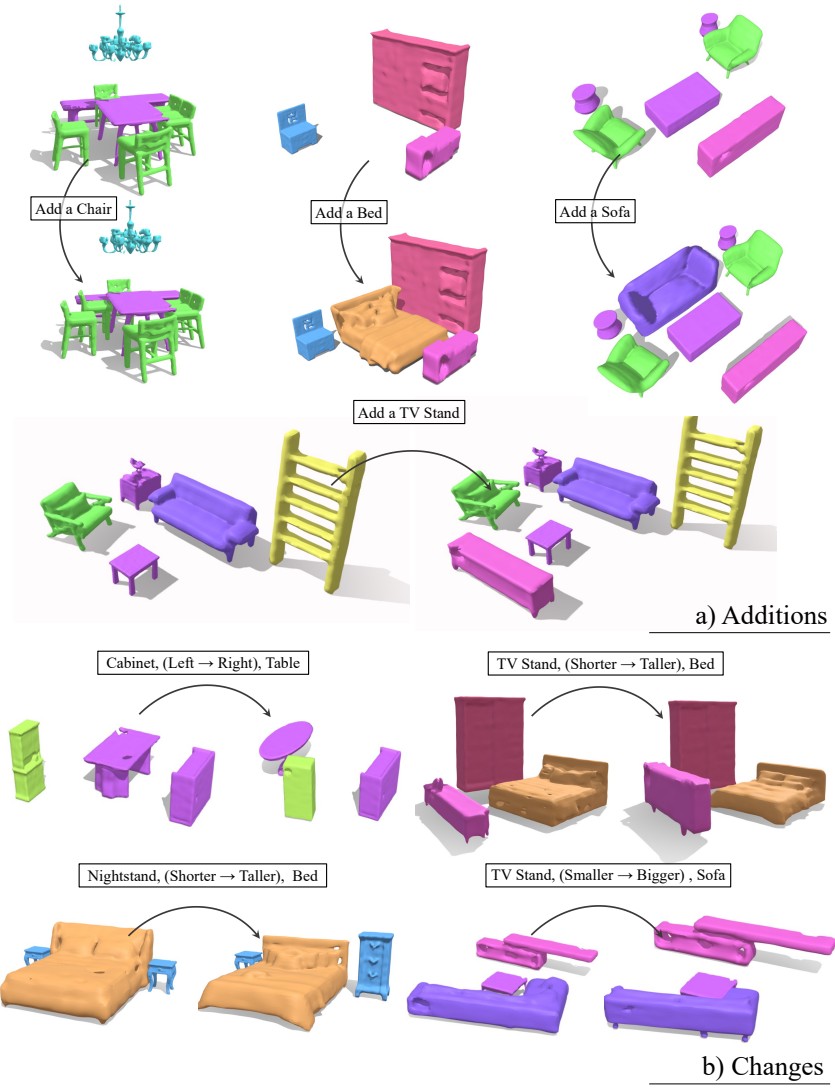

a) Additions

b) Changes

Figure 2: **Scene manipulation results.** Samples from SG-FRONT and 3D-FRONT depicting object additions and relation changes.

| Method | Condition | Corr. & Real. | Style Fit. | Scene graph |
|---|---|---|---|---|
| Layout+txt2shape | Ours | 0.16 | 0.18 | 0.06 |
| Graph-to-3D [3] | Ours | 0.17 | 0.20 | 0.15 |
| Ours | All | 0.83 | 0.80 | 0.87 |

Table 3: **Perceptual study results.** Results over paired study, where the users indicated the preference over A/B scene comparison.

## 3  SG-FRONT Dataset Details

We have annotated the 3D-FRONT dataset [4] with scene graphs to create the SG-FRONT dataset. For annotating the layouts with scene graphs, we follow the previous works 3DSSG [10] and 4D-OR [12] with a semi-automatic annotation approach. The dataset statistics regarding the number of relations and objects per room type are summarized in Figure 5.

Furthermore, the original 3D-FRONT dataset comprises a large variety of scenes, including noise and unrealistic clutter, and previous work used a preprocessed version of this dataset that encompasses three room types (living room, dining room, and bedroom) and removes extremely cluttered scenes,

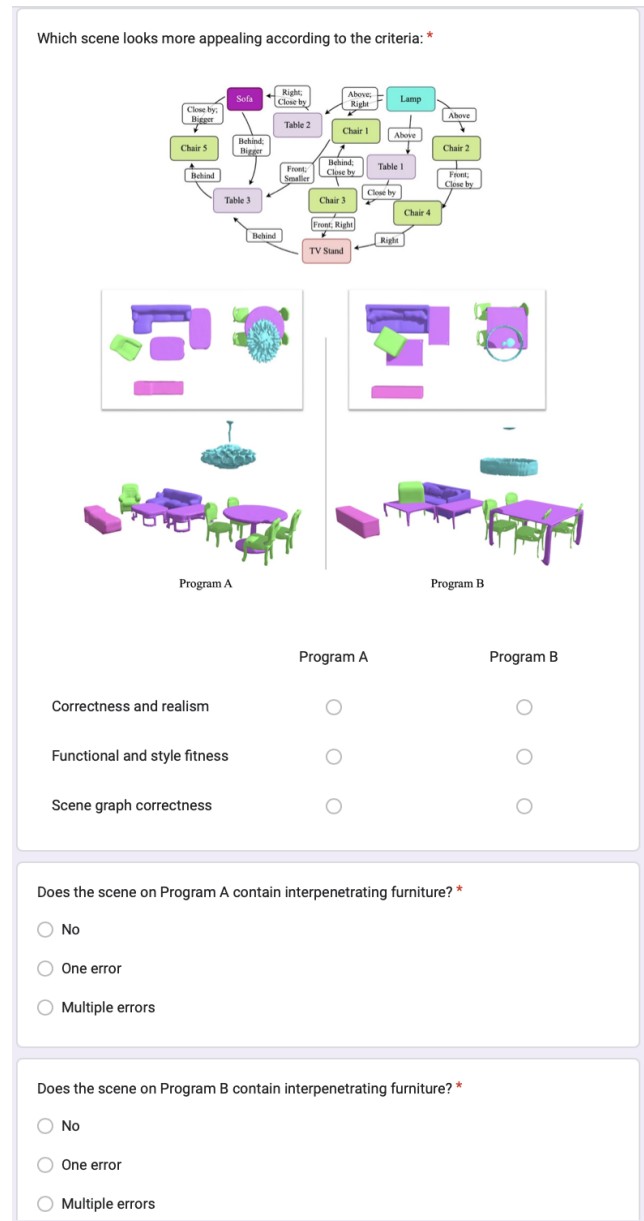

Figure 3: **User interface** for the perceptual user study.

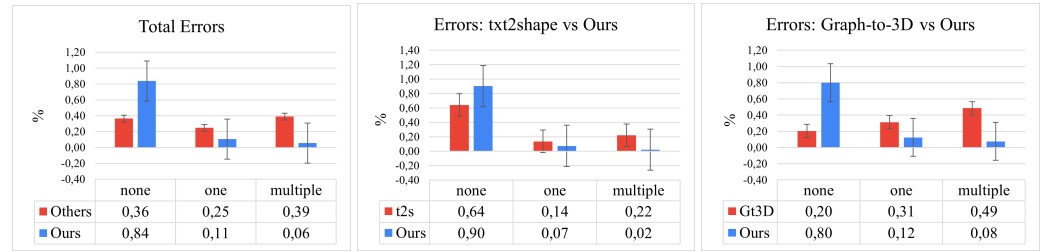

Figure 4: **Error bars** to indicate the number of errors in scenes observed by the user study participants.

similar scenes, or the scenes that contain a high amount of collision. We follow the same preprocessing criteria and train/test splits provided at ATISS [8]. The whole SG-FRONT contains the scene graph

annotations of 4,041 bedrooms, 900 dining rooms, and 813 living rooms, containing 5,754 scenes in total, the same rooms as processed 3D-FRONT. There are 4,233 distinct objects with 45K instance labels in the scenes.

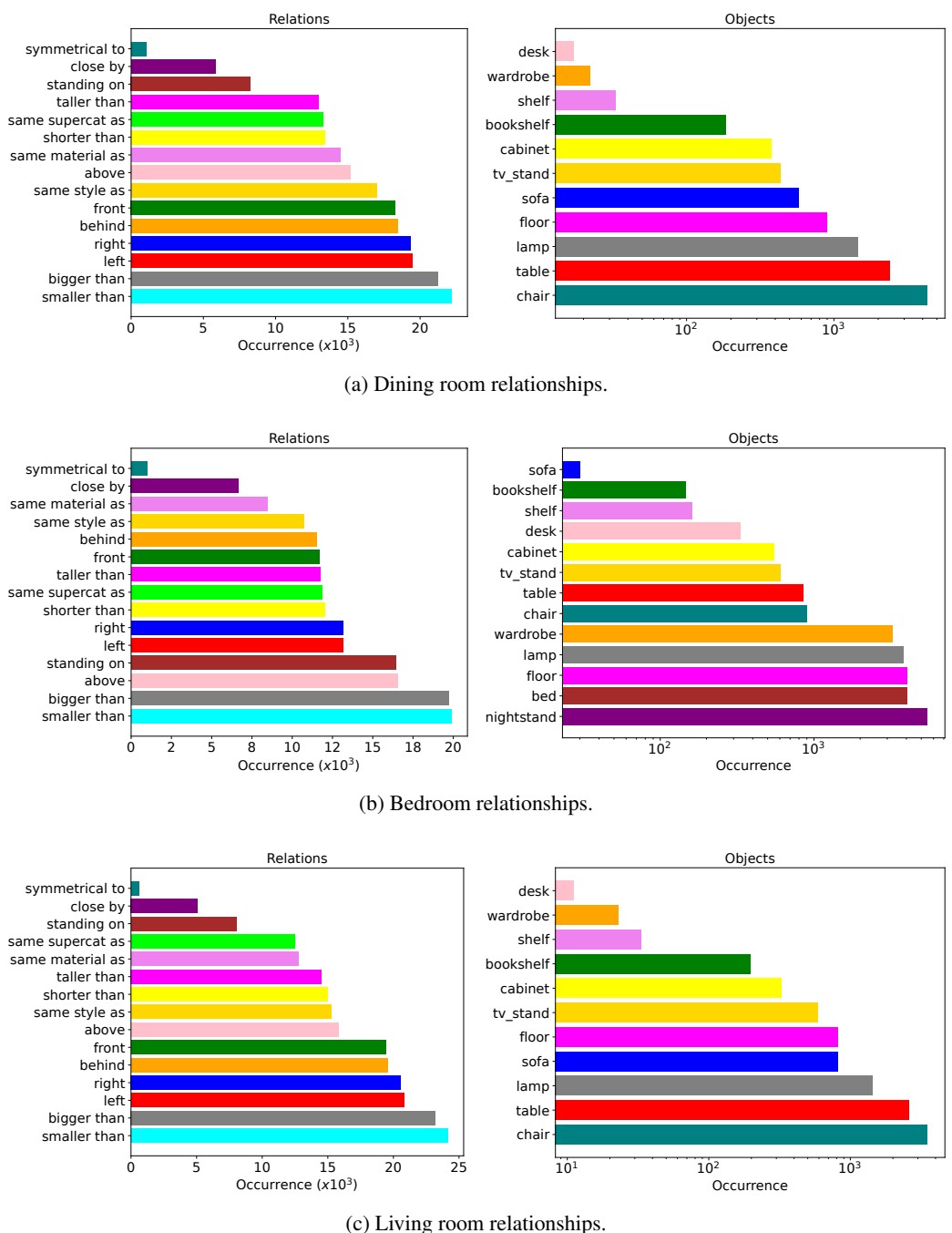

(a) Dining room relationships.

(b) Bedroom relationships.

(c) Living room relationships.

Figure 5: **SG-FRONT dataset statistics** for (a) Dining room, (b) bedroom, and (c) Living room scenes. Relationships are provided on the left, and the object occurrences on the right.

# 4 Results on 3DSSG Dataset

**Discussion over 3RScan versus 3D-FRONT.** The existing scene graph datasets lack high-quality object-level meshes to facilitate our task. One previously introduced such dataset, 3DSSG, is built

upon the 3RScan dataset [9], which consisted of scanned indoor environments from a Tango mobile phone to detect camera relocalization in changing indoor environments. Further, the semantic scene graph annotation, 3DSSG [10], is provided means to alleviate the problem of camera relocalization from 3D scene graphs. Graph-to-3D built their method with this dataset since it was the only available indoor 3D scene graph dataset at the time. Their AtlasNet version could generate scenes in sparse point cloud format, yet, it is inherently inferior to the DeepSDF version in terms of the quality of dense reconstruction. However, since 3RScan cannot be used to train the DeepSDF branch due to its noisiness and incomplete meshes, Graph-to-3D trains on object-level meshes from ShapeNet [1]. Although it can successfully enable the method to proceed, the fragments between 3RScan and ShapeNet prohibit the method from being fully functional and tested correctly. To achieve high-quality scene generation and a fair baseline comparison, we construct a scene graph dataset SG-FRONT based on 3D-FRONT [4]. 3D-FRONT is a large-scale indoor synthetic dataset with professionally designed layouts and a large variety of objects reflecting natural environments. The scenes contain stylistically consistent 3D objects placed according to the choice of interior designers. Compared to 3DSSG, this dataset facilitates 3D scene generation research by providing high-quality watertight 3D object meshes from 3D-FUTURE [5].

**Comparison to baselines.**   For complementary reasons, we provide results on the 3DSSG dataset as well. Since the usage of 3DSSG is not directly possible [3] and would give suboptimal results within our method with the ShapeNet dataset, we evaluate our method and compare it with the previous methods in terms of layout generation. We again use scene graph constraints to measure layouts. In this aspect, we use the original five metrics (`left/right`, `front/behind`, `smaller/larger`, `lower/higher`, and `same as`), which was excluded in previous work [3]. Scene generation results are shown in Table 4, and the scene manipulation results (in addition and change modes) are given in Table 5. In the generation phase, our method has impressive results compared with others, showing that our contextual graph can improve the performance of layout understanding. In manipulation, our method still shows strong results, where `left/right`, `smaller/larger`, `lower/higher`, and `same as` are better than others.

| Method | left / right | front / behind | smaller / larger | lower / higher | same as |
|---|---|---|---|---|---|
| 3D-SLN [6] | 0.74 | 0.69 | 0.77 | 0.85 | **1.00** |
| Graph-to-Box | 0.82 | 0.78 | 0.90 | **0.95** | **1.00** |
| **Ours** w/o SB | **0.90** | **0.84** | **0.98** | **0.95** | **1.00** |

Table 4: **Scene generation results on 3DSSG** in terms of graph constraints (higher is better).

| Method | Mode | left / right | front / behind | smaller / larger | lower / higher | same as |
|---|---|---|---|---|---|---|
| 3D-SLN [6] | | 0.62 | 0.62 | 0.66 | 0.67 | 0.99 |
| Graph-to-Box [3] | change | 0.65 | 0.66 | 0.73 | 0.74 | 0.98 |
| **Ours** w/o SB | | **0.71** | **0.71** | **0.76** | **0.80** | **1.00** |
| 3D-SLN  [6] | | 0.62 | **0.63** | 0.78 | 0.76 | 0.91 |
| Graph-to-Box [3] | addition | 0.63 | 0.61 | 0.93 | 0.80 | 0.86 |
| **Ours** w/o SB | | **0.72** | 0.62 | **0.94** | **0.90** | **1.00** |

Table 5: **Scene manipulation results on 3DSSG** in terms of graph constraints (higher is better). Top: Relationship change mode. Bottom: Node addition mode.

## 5   More Qualitatives on Scene Generation

We show more quantitative results in Figure 6, and 7 to illustrate that our method can achieve realistic generation, higher object-object and scene-object consistency. Generated shapes should be realistic, while Layout+txt2shape can only randomly generate a lamp but cannot consider whether the real size matches the bounding box size in Figure 6, making it stretch too much to be unrealistic. In contrast to these methods, we can achieve every requirement in the scene. For the example of object-object

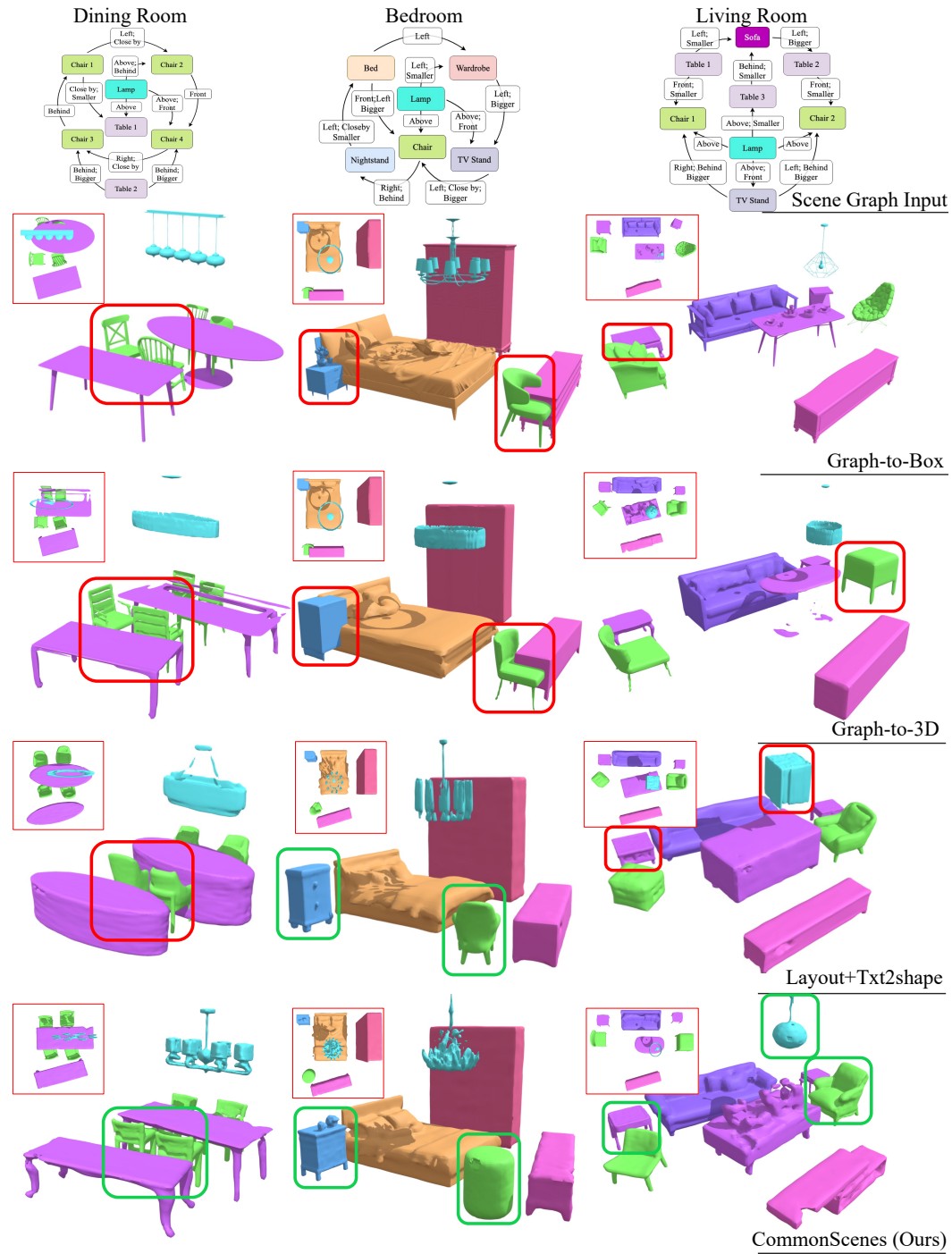

Figure 6: **Additional generation results.** Red rectangles show both the scene-object and object-object inconsistency, while green ones highlight the reasonable and commonsense settings.

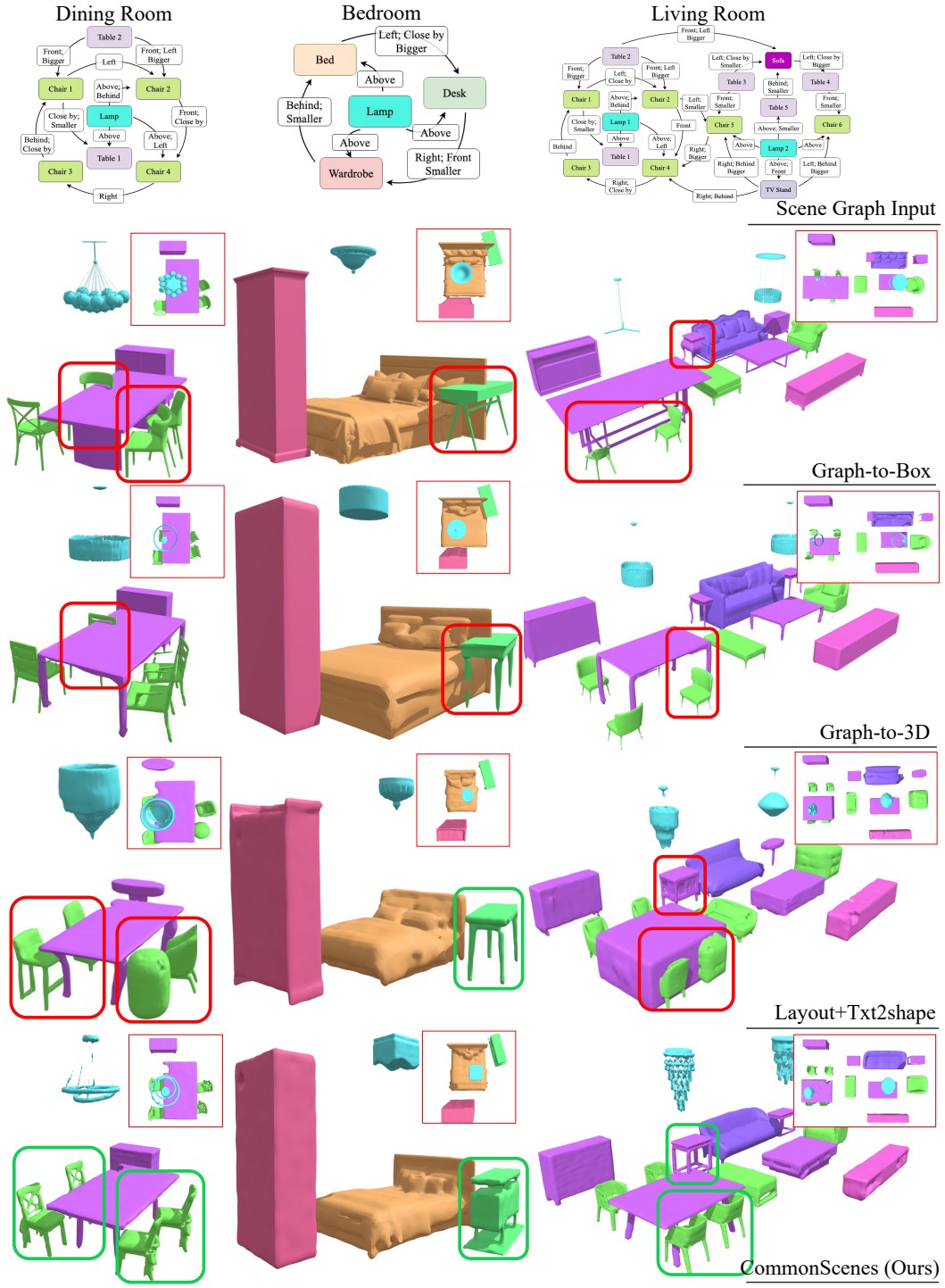

Figure 7: **Additional generation results.** Red rectangles show both the scene-object and object-object inconsistency, while green ones highlight the reasonable and commonsense settings.

consistency in Figure 6, in the dining room, the other three methods cannot generate a suit of dining chairs, while our method can achieve the goal. For the scene-object consistency, in the living room in Figure 7, even though Graph-to-3D can generate consistent chairs, but they are not suitable with the table, e.g., incompatible height and unaligned orientation.

## 6 Discussion

We show more collisions existing some of the scenes in Figure 8. Besides this main limitation, also mentioned in the main paper, we have deliberately excluded texture and material information from our methodology and focused on generating stylistically coherent and controllable 3D scenes. Incorporating additional texture/material details would introduce a new dimension of complexity to the method, as modeling unbounded 3D shapes with materials is not straightforward. Hence, including texture and material information is an exciting avenue for future research.

Regarding ethical considerations, our method does not involve using human or animal subjects, nor does it introduce direct ethical implications. However, being a generative model, it shares ethical concerns commonly associated with other generative models, particularly regarding potential misuse. We are hopeful that the broad impact of generative models will outweigh any negative use cases and that the wider community can leverage this powerful technology for positive advancements.

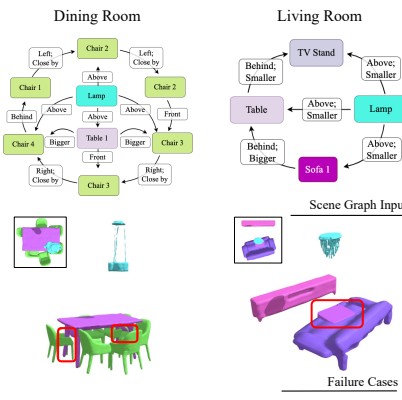

Figure 8: More interpenetrations.

## 7 Additional Training Details

### 7.1 Implementation Details.

**Trainval and test splits.** We train and test all models on SG-FRONT and 3D-FRONT, containing 4,041 bedrooms, 900 dining rooms, and 813 living rooms. The training split contains 3,879 bedrooms, 614 dining rooms, and 544 living rooms, with the rest as the test split.

**Batch sizes.** The two branches use individual batches in terms of the different training objectives. The layout branch uses a scene batch during one training step, containing all bounding boxes in $B_s$ scenes. The shape branch is supposed to take an equal amount of the relation embeddings out of the relation encoder $E_r$. However, this way is prohibited by the limitation of the memory of the GPU. The shape branch instead takes an embedding batch containing $B_o$ embeddings of the counterpart objects sampled by our *Uniformed Sampling* strategy shown in Figure 9. Our training strategy can support sufficient training using little memory storage and data balance. Given all objects (colored balls) in $B_s$ scenes (colored rectangles), we first set the sampling goal as to obtain $\lceil B_o/B_s \rceil$ objects in each scene and collect all objects in each class of $n$ classes in the scene. Take Scene 1 as an example. We divide all $Q$ objects into $n$ classes: $Q = \sum_{i=1}^{n} Q_i, Q_i \geq 0$, where $Q_i$ means the number of objects in the $i$-th class, and similarly treat other scenes. We can achieve class-aware sampling in this way to ensure that every class in each scene experiences the training process. Then we sample an object set $\{q_i | q_i \geq 0, i = 1, \ldots, n\}$ out of $Q$ objects using random sampling, where $\lceil B_o/B_s \rceil = \sum_{i=0}^{n} q_i, i = 1, \ldots, n$. We combine the sampling sets of all scenes and prune objects to $B_o$. Finally, we feed the corresponding relation embeddings into the shape branch. We set $B_s$ as 10 and $B_o$ as 40 in the experiment.

### 7.2 Baseline training.

**Graph-to-3D.** We train the DeepSDF [7] version of Graph-to-3D, as this is the only version that can achieve SDF-based generation. Since DeepSDF requires per-category training, for the shape decoder of Graph-to-3D, we train twelve DeepSDF models. We do not train on "floor", as this category includes no meshes and is a virtual node in scene graphs. We first train DeepSDF models

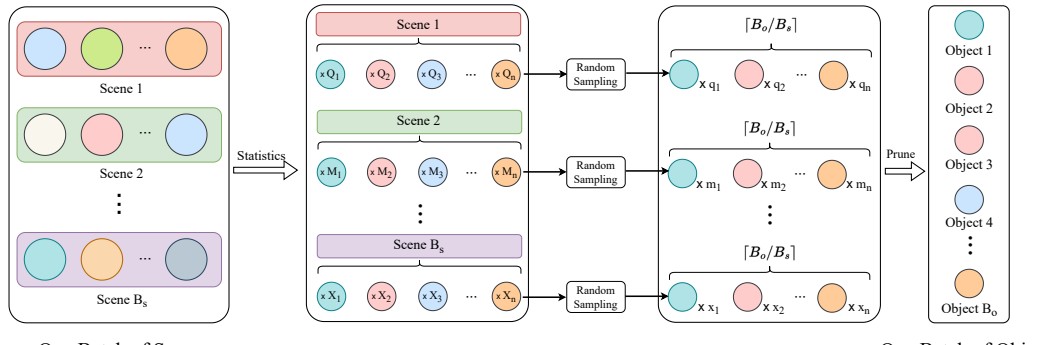

Figure 9: **Uniformed Sampling.** Objects are represented as colored circles, while scenes are colored rectangles.

following the protocol in the original work [7] with 1,500 epochs, with the objects that appear in training scenes of SG-FRONT. After training, we optimize the latent code of each training object and store the embeddings. We then train Graph-to-3D with the latent codes following their public training protocol [3]. At the inference time, we use the predicted latent code directly to generate the 3D shapes and, thereby, with the layout, the entire scene.

**Graph-to-Box.** It is the layout prediction mode of Graph-to-3D without shape prediction. We remove the shape branch and train it using the same settings as in Graph-to-3D.

**3D-SLN.** We follow the implementation and training details provided by authors [6]. We train this baseline for 200 epochs and select the best by the validation accuracy.

**Progressive.** This is a modified baseline upon Graph-to-Box, specifically adding objects one by one in an autoregressive manner [3]. This can be seen as a method to function in manipulation mode. We train it for 200 epochs and select the best by validation accuracy.

**Layout+txt2shape.** We utilize the state-of-the-art model SDFusion [2] as the text-to-shape model and collect all objects in the training scenes for SDFusion to train for 200 epochs using a learning rate of $1e-5$ as proposed in the original paper. Then we train the layout branch solely with the same training settings in our pipeline. Finally, we connect the layout branch with the SDFusion model in series to finish this baseline.

### 7.3 Open source artifacts.

The following open-source artifacts were used in our experiments. We want to thank the respective authors for maintaining an easy-to-use codebase.

- PyTorch3D
- Trimesh
- Open3D
- fid-score
- SDFusion
- DeepSDF
- Graph-to-3D
- ATISS