# OpenReview forum: "CommonScenes: Generating Commonsense 3D Indoor Scenes with Scene Graph Diffusion"
_NeurIPS.cc/2023/Conference — NeurIPS 2023 poster_

### Official Review · Reviewer_zttF · 2023-06-25

**Soundness:** 3 good
**Presentation:** 2 fair
**Contribution:** 3 good
**Rating:** 7
**Confidence:** 3

**Summary:**

The paper looks at the problem of learning a generative model for sampling 3D environments from a description based on a graph and natural language. The graph does correspond to objects that participate in the scene, and textual descriptions attached to nodes and edges further specify the scene. Textual descriptions are encoded using CLIP. A variational auto encoder conditioned on these inputs is trained to generate the bounding boxes of the 3D objects. A latent diffusion model is trained in parallel to generate the 3D shape of the object in each bounding box. The existing FRONT-3D dataset is augmented with textual descriptions for training this model. Experiments show the advantages and limitations compared to ablations and prior works.

**Strengths:**

* Scene generation is an important problem and this paper addresses it with a responsible model using modern components in a competent manner.

* The paper contributes a large amount of additional labels for 3D-FRONT.

**Weaknesses:**

* The  presentation, especially of the technical parts, can be improved. It was difficult to understand how the layout auto-encoder is setup. It would be useful to give a high-level overview first of where the method is trying to achieve (e.g., encoder Z = E(P, O, B), decoder \hat B = E(P, O, Z), Z Gaussian). Section 4.1 presents only half of the story, and we need to get to section 4.2 before the auto-encoder materialises. Line 143 literally states the the encoder is trained by minimising Eq. (4), which is clearly not enough -- Eq. (5) is also needed.

* You man need to revisit the formalism at Ines 90 to 99. $c_i^{node}$ and the same terms for the edges are very poorly defined. Furthermore, it also seems that you can attach more than one such attributes to each node or edge. The formalism does not allow for that.

* Some aspects of the model are unclear. See questions below.



**Questions:**

* Please clarify how the vectors o in Figure 2 and line 99 are obtained. The paper only states that they are "learnable".

* The two shape and layout branches are said to be trained jointly (Section 4.4). However, the learning dynamics of auto encoder and diffusion processes (one per object) seem to me to be very different. Is it really that trivial to mix them?

* Line 172: the shape diffusion model connects to the *entire* graph, and so to all objects, via cross-attention. How does each instantiation of the diffusion process know *which* of the several objects it is meant to generate?

* Line 214: I don't know what it means to share the same CAD model source. Nor I do understand why it is is sensible to compare a generated sample to a specific CAD instance -- I would not expect the two to match.




**Limitations:**

There is sufficient discussion of limitations, but no discussion of ethics. The latter, however, is not particularly relevant for this paper. It may be worth at least discussing the licensing of 3D-FRONT.

---

> ### Author Rebuttal · Authors · 2023-08-09
>
> ## 1. Overview paragraph on optimization
> Thanks for the useful advice! We now extend our overview section (L111-116) by explicitly cross-referencing figures and the subsections. Further, we rephrase the sentence in L143 as: *“We guide the distribution of embedding space through employing the KL loss.”*. Note that in Section 4.4, we already explained that we use the combination of all these losses to optimize the network, but we will clarify the necessity of this up-front by cross reference.
> ## 2. The formalism in the preliminary
> We have fixed L93 thusly to clarify the formalism:
> *Each vertex $v_{i}$ is categorized  through an object class $c^{node}_{i} \in \mathcal{C}^{node}$, where $\mathcal{C}^{node}$ denotes the set of object classes.* Note that in terms of notations, we follow the prior work [58].
> ## 3. Clarification about vectors $o$
> The vectors $o$ in Figure 2 and L99 are learnable object embeddings, where each object class is identified with one fixed-size embedding initiated randomly and optimized over the training. It is in the same vein as how Large Language Models use learnable tokens per word in a dictionary.
> ## 4. Joint training of two branches
> Joint training can be achieved. We train the layout VAE and the latent shape-diffusion branches together with the combination of losses (Eq. 7). It was indeed not trivial to mix them, which is why we discussed potential implications in the Supp. Mat. (L155-170), and we ensured synchronized training between the two branches with a uniform sampling (Supp. Mat. Figure 9).
> ## 5. Cross-attention feature conditioning
> The shape branch is conditioned on per-node relation embedding via cross-attention to generate per-node shapes, not the entire graph. We then run the diffusion for every node in the graph to populate the scene. We will clarify this in the final version.
> ## 6. CAD model comparison
> The intention is to measure the generation consistency in the dining rooms, where dining chairs usually appear together in a suit and dining tables as well (p.8 Figure 8). To collect the consistency ground truth, for example, in one dining room, we collect the chairs that are decorated using the same textured CAD model. This shows us which chairs should be in a suit after the generation. We then calculate the CD between these chairs, the smaller the CD the better. This metric is mainly used to test object-object consistency (p.8 Table 2).
> ## 7. License discussion
> We have discussed the ethics in Supp. Mat. L145-149. In terms of licensing, we will follow CC BY-NC-SA.

---

> > ### Comment · Reviewer_zttF · 2023-08-11
> >
> > I thank the authors for their response and encourage them to incorporate these clarifications in the final version of the paper.

---

### Official Review · Reviewer_NT6V · 2023-07-06

**Soundness:** 3 good
**Presentation:** 3 good
**Contribution:** 3 good
**Rating:** 6
**Confidence:** 3

**Summary:**

Summary: This paper presents a generative model to generate 3D scenes from scene graphs. Their model is fully generative without the need of any shape database or embeddings. The 3D scenes generation model is pipelined into finding the scene layout and the construction the shapes of the nodes using a diffusion model. The authors constructed a scene graph dataset SG-FRONT from 3D-FRONT dataset and trained end to end to generate 3D scenes from the scene graph input. The paper reported qualitative, quantitative results with comparison to other sota models on the SG-FRONT dataset.

**Strengths:**

1. the paper claims to have curated a dataset(enriching an existing dataset with scene graph annotations) for 3D scenes construction from scene graphs.
2. Codes and dataset will be publicly available
3. The generated 3D scenes and the quantitative numbers show potential of the scene graph based approach for 3D scenes reconstruction using a diffusion model

**Weaknesses:**

1. In section 6, Compared Baselines, 'a fully generative method a text-to-shape generation model that follows a layout generation' does not have citations. Is it something the authors modeled for experimentation?
2. In Table 1, need to specify which of the methods use fully generative approach with text only. Also, calrifying in the caption of table 1 on how you segmented table 1 into two main rows and what is 'Ours 'w/o SB''
3. Sharing more details on how the authors curated the dataset from 3D-FRONT in the main paper might be helpful for the readers
4. Some possible typos:
line 47 : 'cues and fine local inter-object relationships.'
Line 65 and 66: Incomplete lines : 'Quickly after this progress, the 3D [53, 1, 58, 26], dynamic [44], robotic
66 grounding [19, 45], spatio-temporal 4D [66], and controllable scene synthesis [29, 65, 54, 36, 13].'

**Questions:**

1. In section 6, Compared Baselines, 'a fully generative method a text-to-shape generation model that follows a layout generation' does not have citations. Is it something the authors modeled for experimentation?

**Limitations:**

no negative societal impact.
No limitation addressed.
All the 3D scene generation is done on synthetic dataset. How would the whole model perform on the real scene?

---

> ### Author Rebuttal · Authors · 2023-08-09
>
> ## 1. The baseline based on the text-to-shape model
> It is a method we modeled for experimentation. The motivation is introduced in L36-40, the main paper. We have also further explained the implementation of this baseline in Supp. Mat. Section 7 in detail. This baseline consists of a layout generator (the same as ours) and a single text-to-shape generator (SDFusion [8]). Given a scene graph describing the target scene, this baseline uses the layout branch to generate the bounding boxes, while concurrently feeding the individual category name of each node in the scene graph into the text-to-shape generator to generate object shapes, e.g., using the word “bed” with SDFusion to generate a plausible bed. Finally, the entire scene is synthesized by populating each shape within its corresponding sounding box. Here, we train the layout branch using the same settings as ours and train the text-to-shape generator following SDFusion.
> ## 2. Clarification on Table 1
> Among the baselines evaluated, only “Layout+txt2shape” is the “a fully generative approach with text only”. Two main rows are separated with respect to the reliance on an external shape database for retrieval. “Ours w/o SB” refers to ours without the shape branch. We will clarify these aspects in the main paper to ensure that readers can fully understand the distinctions.
> ## 3. More dataset collection details in the main paper
> We plan to extend Section 3 of the main manuscript through the details from Supp. Mat. Section 3 Dataset Details (L82-86) upon acceptance. In core, we adopt a semi-automatic approach for detecting spatial layout through bounding box extensions followed by human annotator inspection (as in prior work 3DSSG [53], 4D-OR [66]). For annotating semantic-level edges, we resort to 3D-FRONT [15] annotations where we extract the object-level features (e.g., “same material as”, “same style as”).
> ## 4. Limitations, societal impact, and real-world performance
> In the Supp. Mat. Sec. 6 Discussion and Limitations, we addressed the negative societal impact (L145-149) as well as the limitations (L132-144). We plan to carry the essential parts to the main manuscript.
> Furthermore, for real-scene performance, we also provided a discussion and analysis in Supp. Mat. Section 4 Results on 3DSSG dataset [53].

---

> > ### Comment · Reviewer_NT6V · 2023-08-20
> >
> > I appreciate the response from the authors. All my concerns have been addressed. Also, thanks to the author for pointing out the limitation stated in the review, which I missed from the supplementary materials.

---

### Official Review · Reviewer_ANPx · 2023-07-09

**Soundness:** 3 good
**Presentation:** 3 good
**Contribution:** 3 good
**Rating:** 6
**Confidence:** 5

**Summary:**

This paper addresses the task of controllable scene synthesis of indoor rooms, conditioned on a semantic scene graph that captures spatial, style and support relationships between objects in a scene. In particular, they introduce CommonScenes, a generative model capable of converting scene graphs into 3D scenes using diffusion models. Given a scene graph of the scene to be generated, they first enhance it using CLIP features computed on the inter-object relationships, as well as using the ground-truth bounding box annotations. Next, they leverage a triplet-GCN based relation network to propagate information among the objects in the scene and produce the Box-enhanced Contextual Graph (BCG). Given a BCG, they then utilize a dual-branch network that generates the final scene. The first branch (layout branch) is another triplet-GCN that generates 3D layout predictions and the second branch (shape branch) generates shapes for each node represented as SDF. The shape branch is simply a latent diffusion model. During training, CommonScenes considers supervision in the form of per-object bounding-box annotations (size, location, rotation), as well as per-object 3D meshes in the form of SDFs. To evaluate their model, the authors enrich the 3D-FRONT dataset with scene graph labels, by annotating per-object spatial, style and support relationships. The authors compare their model to several baselines and show that their approach yields more realistic scenes in terms of FID and KID scores.

Overall, I think this is a nice work that alleviates the need for relying on a library of assets to replace the generated bounding boxes with 3D objects. The proposed architecture is novel and seems to be able to consistently produce plausible scenes. That being said I think that the proposed pipeline is quite complex, as it consists of multiple submodules, and it requires conditioning in the form of semantic graphs that are relatively hard to acquire, hence the authors had to enrich 3D-FRONT. However, since the authors show that their model outperforms prior research I am in favor of accepting this paper.


**Strengths:**

1. To the best of my knowledge, the proposed model is novel and the authors clearly demonstrate that their model consistently produces plausible scenes. I think that this work is an important step towards alleviating the need for large libraries of assets when generating a novel scenes. Relying on such libraries naturally restricts the diversity of the generated scenes to the diversity of the objects in the library. Although, the proposed model is a relatively complex, I believe it is a valuable work that could potential inspire other works in this direction.

2. I think that the development of the SG-FRONT dataset, that extends 3D-FRONT with scene graph labels is an important contribution of this work that can potentially facilitate other research projects, therefore I strongly encourage the authors to make this dataset variant available upon the paper's acceptance.

3. I really appreciated the supplementary video that the authors provided. I particularly liked the intuitive explanation of the proposed method as well as the additional results. In the future, I hope that more authors will provide such high quality supplementary videos alongside with the paper submission.

4. Although the proposed model is quite complex, I think the paper is nicely written and easy to follow. I really liked the provided figures that provide a more high-level pictorial overview of the various components of the proposed pipeline.


**Weaknesses:**

1. The main weakness of this work is that it is relatively complex, as it consists of multiple sub-modules. One thing that was not 100% clear from the text was whether they perform a two-stage training, namely first produce the BCG, as discussed in Sec 4.1 and then jointly train the shape and layout branch of CommonScenes, as discussed in Sec 4.2. It might be good to clarify this for the final version of the paper. In addition, for the Shape Decoding module (L164-179), I am not 100% sure whether the authors use some sort of class conditioning to train one LDM for all object types or they use separately trained LDMs per-object class. This might be good to clarify.

2. Although, the paper focuses on scene synthesis, I think that an important evaluation that is missing is measuring the quality of the generated shapes. In particular, the authors could generate a couple of rooms and then take the beds, chairs, nightstands etc. and compute the COV and MMD of the generated objects w.r.t the ground-truth objects. Although, object generation is not the main focus of this work, I think this analysis would be valuable, since this is an integral component of the proposed model. I noticed that the authors tried to do this type of analysis in Sec. 1.1 in their supplementary, but I found it a bit weird that they report Chamfer Distance, instead of MMD/COV. Is there any reason for this?


**Questions:**

1. For the Ablations in Sec 6.3, for the model variant that is trained without context, do the authors omit both the CLIP features, as well as the bound-box features or only the former. I think that they should be ablated separately. Personally, I am not 100% convinced about the significance of the CLIP features. Can the authors please clarify?

2. I am wondering whether the authors tried to condition their scene generation also on the floor plan together with the semantic scene graph? Unless I am missing out something, this can be easily done e.g. by extracting features from the floor plan and concatenating with the per-object features while created the BCG? Do the authors think that something as simple as that would work?

3. One thing is that a bit unclear to me is whether the first component of the pipeline that generates the BCG using the triplet-GCN is really necessary. Have the authors tried to directly use the scene graph enhanced with the CLIP features and the bounding box features? This might be an interesting ablation to provide for the final version of the paper.

4. One concurrent work that I think the authors should add in their reference list is the Learning 3D Scene Priors with 2D Supervision, CVPR 2023. I think this work could also be an interesting baseline as they can generate both the scene layout as well as the 3D shapes

5. A minor comment/suggestion: In the caption of Figure 1 in the supplementary material the authors state "shows a huge diversity". I think huge might be slightly overclaiming. It might be better to tone it down a bit.


**Limitations:**

The authors discuss the limitations of their work and the potential negative impacts in the society in their supplementary material. I think they could have also provided some qualitative examples of the failure case of their model to help the reader better understand the limits of this work.

---

> ### Author Rebuttal · Authors · 2023-08-09
>
> ## 1. Training details
> We perform one-stage training. As mentioned in the L129, the BCG is created and encoded on-the-fly by the contextual encoder $E_c$. We agree that the overview needs to indicate one-stage training clearly. As for LDM, we train a single LDM for all object types conditioned on the learned relation embeddings. We will clarify both aspects in the final version.
> ## 2. More object-level evaluation
> Since our objective is scene generation, we use FID/KID as the main metrics for evaluating the scene-level generation quality. **On the other hand, we reported the CD as a metric to evaluate generation diversity, following the previous state-of-the-art method Graph-to-3D to enable a fair comparison**  (more details in Supp. Mat. L16-17).
> As requested, we report the MMD (x0.01) and COV (%) for evaluating per-object generation. We collect ground truth objects in each category within the test set and use the evaluation script from PointFlow [72].
>
> **Table 1.** MMD(↓) comparison
>
> |      Method     |   Bed  | Nightstand | Wardrobe |  Chair |  Table | Cabinet |  Lamp  |  Shelf |  Sofa  | TV stand |
> |:---------------:|:------:|:----------:|:--------:|:------:|:------:|:-------:|:------:|:------:|:------:|:--------:|
> | Graph-to-3D | 1.56 | 3.91     | 1.66   | 2.68 | 5.77 | 3.67  | 6.53 | 6.66 | 1.30 | 1.08 |
> | Ours    | **0.49** | **0.92**     | **0.54**   | **0.99** | **1.91** | **0.96**  | **1.50** | **2.73** | **0.57** | **0.29** |
>
> **Table 2.** COV(↑) comparison
>
> |      Method     |   Bed  | Nightstand | Wardrobe |  Chair |  Table | Cabinet |  Lamp  |  Shelf |  Sofa  | TV stand |
> |:---------------:|:------:|:----------:|:--------:|:------:|:------:|:-------:|:------:|:------:|:------:|:--------:|
> | Graph-to-3D | 4.32 | 1.42 | 5.04   | 6.90 | 6.03 | 3.45  | 2.59 | 13.33 | 0.86 | 1.86 |
> | Ours    | **24.07** | **24.17** | **26.62** | **26.72** | **40.52** | **28.45**  | **36.21** | **40.00** | **28.45** | **33.62** |
>
> As shown in Tables 1 and 2, our method shows better performance in both MMD and COV, which highlights the object-level shape generation ability of CommonScenes.
> We also calculate 1-nearest neighbor accuracy (1-NNA, %), which directly measures distributional similarity on diversity and quality. This measurement has motivated us to include it in this rebuttal as well for reviewers’ and readers’ reference. The closer the 1-NNA is to 50%, the better the shape distribution is captured.
>
> **Table 3.** 1-NNA(↓) comparison
>
> |      Method      |     Bed    | Nightstand |  Wardrobe  |    Chair   |    Table   |   Cabinet  |    Lamp    |    Shelf   |    Sofa    |  TV stand  |
> |:----------------:|:----------:|:----------:|:----------:|:----------:|:----------:|:----------:|:----------:|:----------:|:----------:|:----------:|
> | Graph-to-3D   | 98.15     | 99.76     | 98.20     | 97.84     | 98.28     | 98.71     | 99.14     | 93.33     | 99.14     | 99.57     |
> | Ours | **85.49** | **95.26** | **88.13** | **86.21** | **75.00** | **80.17** | **71.55** | **66.67** | **85.34** | **78.88** |
>
> It can be observed that our method surpasses Graph-to-3D in the evaluation of distributional similarity. Coupled with the results in Tables 1 and 2, CommonScenes exhibits more plausible object-level generation than the previous state-of-the-art. We will extend our Supp. Mat. with these additional experiments. Please also check our summarized table in the attached PDF.
> ## 3. Ablations on the CLIP features
> Exactly as the reviewer indicated, we ablate the CLIP features only in Section 6.3, *“Ours w/o context”*. By omitting the features, the BCG degrades to a box-enhanced scene graph, lacking strong semantic cues to guide the shape and consistent layout generation. Table 5 shows this is vital for the quality of 3D scene generation results, both in terms of synthesis quality (FID/KID) and the relation correctness (mSG). Additionally, compared to layout generation in Tables 1, 3, and 4, our method without a shape branch (*"Ours w/o SB"*) highlights the effectiveness of BCG. On the other hand, the ground truth bounding box parameterization process cannot be ablated since it is essential to the VAE modeling, which also serves as supervision labels for the layout.
> ## 4. BCG encoding
> We did not use the triplet-GCN to "generate" a BCG, Instead the BCG is encoded by the triplet-GCN-based $E_c$ during the training, and we actually followed the suggested approach as mentioned in L128-131. For even more clarity, we will rephrase the relevant part in the final version.
> ## 5. Floor plan involvement
> No, we have not tried that. Our motivation is to use as simple as possible conditions, i.e., only objects and their relationships, to condition a scene generation. However, the approach mentioned by the reviewer may mean removing the floor node since it would be used in enhancing the per-object features, i.e., by concatenation. Instead, replacing the floor feature from CLIP with a floor plan embedding and treating it as a node can potentially generate user-defined floor shapes. We will investigate this direction as our further work.
> ## 6. A related paper from CVPR 2023
> Indeed this is a relevant work that learns 3D shape priors from 2D images and generates shapes and layouts from a hypersphere space. We now cite this paper published after our work's submission. However, it is worth noting that this method is trained and conditioned on RGB-related information, e.g., 2D bounding boxes and masks. Therefore, a direct comparison between our method and this would be inappropriate since our inputs are in different modalities, where we use the form of a scene graph.
> ## 7. Qualitatives for limitations
> We will append Supp. Mat. Section 6 Discussion and Limitations with illustrations. We provide some examples of these in the attached PDF.
>
> [72] Yang et al. "Pointflow: 3d point cloud generation with continuous normalizing flows." ICCV 2019.
>
> [73] Nie et al. "Learning 3D Scene Priors with 2D Supervision". CVPR 2023.

---

> > ### Comment · Reviewer_ANPx · 2023-08-21
> > **Rebuttal Acknowledgement**
> >
> > I would like to thank the authors for taking the time to address my questions and concerns. As I already mentioned in my review I think this is an interesting work that alleviates the need for having a library of 3D assets, moreover as the authors show that their approach outperforms prior approaches, I think that this paper should be accepted. That being said, I would like to urge the authors to incorporate the additional experiments provided in the rebuttal period in the final version of their paper.

---

### Official Review · Reviewer_Wi3K · 2023-07-15

**Soundness:** 2 fair
**Presentation:** 2 fair
**Contribution:** 2 fair
**Rating:** 4
**Confidence:** 4

**Summary:**

Gist:
The paper presents a framework, called CommonScenes, for generating 3D indoor scenes given scene graphs as inputs. CommonScenes is a dual-branch framework where one branch generates the scene layout using a VAE and the second one generates what the authors call "compatible" 3D shapes using latent diffusion. The claim is that having this second branch (where compatible 3D shapes are generated for populating the generated layout from the first branch) allows capturing global scene-object and local inter-object relationships, something that prior works cannot capture (I am not convinced about this, but more on this later).

The generated scenes can be manipulated by editing the input scene graph, as well as sampling noise in the diffusion process.

The paper also constructs a scene graph dataset using an off-the-shelf 3D scene dataset.

Dataset Used:
3D-FRONT is the base dataset used, which is augmented with scene graph labels and this augmented dataset is termed in the paper as "SG-FRONT" dataset.

Training Mechanism:
Supervised in the form of triplet network setting and latent diffusion models


Evaluation Metrics:
To measure the fidelity and diversity of generated scenes, FID, KID scores at 256x256 pixel resolution between the top-down rendering of the generated and real scenes is used.

To measure shape diversity, each scene is generated 10 times, and the changes in shapes is evaluated using CD.


Baselines and Comparisons:
Three kinds of baselines are compared against:

1) First, a retireval-based method, namely, 3D-SLN from CVPR 2020

2) Second, a semi-generative SOTA method, Graph-to-3D from ICCV 2021, and

3) Third, a text-to-shape generation methods that follows layout generation (this is not cited, so I am not informed by the paper if this was implemented by the authors on their own or if any specific algorithm was re-implemented)



**Strengths:**

+ Conceptualizing layout generation using graphs is a nice concept, although this is not the first time it has been addressed. A structured input modality gives rise to many applications, such as scene editing and modification, as demonstrated in the paper.

+ As seen from Figure 4, the proposed method seems to produce plausible outputs given a scene graph as input. This is also validated quantitatively, although I would only consider Table 1 to be more representative of such quantification than other tables.

**Weaknesses:**

- Not really a concern but this is something that people will find about this paper: the paper is trying to do too many things at once. While this may also be a positive aspect in the era of today's models, a reader cannot clearly discern what design aspect leads to layout improvement and what leads to shape improvements. One may even argue: why not use the shape generation scheme employed as an independent approach and submit a paper if enough novelty exists? You get my point.


- L96-97: How are the edge predicates (like spatial relations "left/right", "closeby" etc.) obtained? Is the dataset manually annotated with semantic scene graph information? If that is the case, then, the problem formulation is weak. What would have been interesting is to automatically extract meaningful semantic scene graphs (especially that ground spatial relations to a reasonable extent) and then use these graphs to generate a 3D scene.

- L3-6: I do not understand the message in these lines. Do you mean to say that existing methods use retrieval-based mechanisms to populate the generated layout, because of which scene-object and object-object relationships are inconsistent? It is not true. So, first, I think it is important to rephrase this sentence. It is conveying an altogether different meaning.

- There is mention of a triplet graph network, triplet-GCN in L46, 129, 148, 159. However, there is not much detail about how the positive and negative examples to train this triplet network are obtained. This is quite important since the training data plays a key role in obtaining meaningful and discriminative embedding spaces in the context of contrastive learning setups.

- I fail to understand why the initial node embeddings, c_i^{node}, which are obtained from one-hot semantic encodings (as per info in L93), should be passed through a pre-trained and frozen CLIP text encoder. It makes sense to pass the initial __edge__ embeddings through it as spatial information needs to be captured and the CLIP text encoder does a good job of mapping the initial English text to a meaningful embedding space. But I cannot understand why the node embeddings c_i^{node} need to be passed through the text encoder from CLIP.

- L37: It is not trivial to obtain text information from input scene graphs. This alternative solution is not so straightforward, unlike what is mentioned in this line.

- L65-66, the sentence is incomplete

- L87-88: LEGO-Net from CVPR 2023 is the first work to leverage diffusion models for scene generation. Even though LEGO-Net is designed for scene rearrangement, I would still place it in the generative model category since it is inherently doing denoising to provide a plausible output. So, these two lines are not correct.

- Figure 4: Again, my question is how are the input scene graphs obtained? I am interested in knowing how the spatial relations on the graph edges are obtained. If this is done in a heuristic manner, it is prone to errors, and I do not think this is trivial to obtain in the presence of multiple objects. Obtaining such spatial relations is a challenge, as widely acknowledged in the community (see pre-LLM works on 3D indoor scene generation using language/text input, such as from Chang 2014, 2015, Ma 2018).

- There are quite a few typos and punctuation errors in the paper (one such example is pointed out below). Need to be corrected.
One Missing period/full-stop symbol in line 24.

**Questions:**

Please see the Weaknesses section above

**Limitations:**

The paper does not discuss the limitations of the proposed approach.

There exist many questions (please see the Weaknesses section above) that can critically limit the application of the proposed approach, starting from the way the input scene graphs are obtained. At the least, a discussion on how this work can address or alleviate such challenges using additional processing should have been discussed.

---

> ### Author Rebuttal · Authors · 2023-08-09
>
> ## 1. Text-to-shape baseline
> We have explained this baseline, named Layout+txt2shape", more in detail in Supp. Mat. Section 7. This baseline solely considers the shape generation from text input, which we establish through the text-to-shape SDFusion [8].
> ## 2. Complexity, contribution, and motivation
> Notably, our main contribution is the proposal of a method to enable a fully generative model of the entire 3D scene from scene graphs, encompassing its layout and 3D geometries holistically. In our experiments, we systematically show that using only layout and retrieval-based shapes (3D-SLN [31]) underperforms and that a semi-generative shape model of Graph-to-3D [13] also fares poorly (p.8, Table 1). We show that it is possible to generate shapes together with scene layouts in the proposed framework. Contextual information improves the coarse inter-object relationships (p. 9, Table 5 row1 vs. row 4), and the GCN further propagates information among objects and learning global cues, and finer local-object relationships (p.9, Table 5 row 2 vs. row 4). Besides, the optimization from the diffusion-based shape branch not only assists layout generation during training but also brings better shape generation quality than previous work (p. 7 Figure 4).
> ## 3. Dataset annotation and scene graph sources
> We obtain the edge predicates (e.g., "left/right", "front/behind", “above/standing on'') in a semi-automatic manner, similar to prior work (3DSSG [53], 4D-OR [66]), as explained in the main paper L190-192, and in Supp. Mat. Section 3 L75-78. Essentially, we use three methods: 1) For spatial relations (e.g., "left/right", "bigger/smaller"), we initiate the process by applying relationship checks on the bounding box extents, followed by collision checks; 2) For support relations (e.g., "standing on", "above"), we apply a set of thresholds for each object type identified by human annotators; 3) For stylistic cues, we refer to 3D-FRONT object annotations [15] to identify meaningful information, generating semantic edge labels. Since we use the synthetic and high-quality 3D-FRONT [15], any potential errors occur systematically, in contrast to real scenes obtained by depth cameras [66,67]. These can then be easily corrected.
>
> The input scene graphs in Figure 4 are the annotated scene graphs from the test scenes of 3D-FRONT. For general acquisition of scene graphs, there are various ways, such as from text (Chang et al. [68, 69], Ma et al. [32]), single image (Chen et al. [70], Dhamo et al. [12]), or video (Wu et al. [58]). However, our work aligns with 3D-SLN [31] and Graph-to-3D [13] in using the scene graph to enable controllable scene analysis. Therefore, our scene graphs are obtained in a manner akin to previous approaches [13]. We recognize the challenge in this task and have added the related work [68,69], to our paper, further discussing this aspect in the Supp. Mat.
> ## 4. The explanation and methodology of retrieval-based methods
> Retrieval-based methods, like 3D-SLN [31], Graph-to-Box [13] and ATISS [39], typically determine retrievals based on the sizes of bounding boxes, as shown in Figure 1. In this case, even a slight variation in the estimated bounding box size can lead to significant and undesired variations in the generated shapes, resulting in the generation of inconsistent scenes (p.8, Table 2). In contrast, our generated shapes and appearances are conditioned on the relations among the objects. This leads to more consistent shape generation (p.8 Figure 5, L244-248).
>
> ## 5. The meaning of triplet-GCN
> In this context, the triplet graph network uses a triplet of “subject-predicate-object” [31]. We don’t use any contrastive learning or triplet loss. We will clarify this aspect for readers more familiar with triplet loss concepts.
>
> ## 6. CLIP embeddings for graph nodes
> We do not pass one-hot embeddings to the CLIP text encoder. Instead, we input the object class names for nodes (e.g., “Bed” and “Table”), as well as the edge class names (e.g., “left”, “right”) as illustrated in Figure 2, exactly because of the intuition explained by the reviewer.
>
> ## 5. Text information acquisition in the scene graph
> In L36-39, we explained that simply replacing the shape retrieval baseline with a text-to-shape generator, where the text is acquired from the scene graph, does not yield good results (i.e., the baseline "Layout+txt2shape"). **The textual input from the scene graph is obtained by means of class names**. We will add a cross-reference to this experiment on L39 to prevent misunderstanding.
>
> ## 8. LEGO-Net from CVPR 2023
> We thank the reviewer for pointing out the very recent work [71]. We will cite this work and refine our statement in related work accordingly.
>
> ## 9. Limitation Discussion
> We discussed the limitations in the Supp. Mat., on Section Discussion and Limitations L131-149. We will carry the essential parts to the main manuscript.
>
> [67] Wald et al. "Rio: 3d object instance re-localization in changing indoor environments." ICCV 2019.
>
> [68] Chang et al. "Learning spatial knowledge for text to 3D scene generation." EMNLP 2014.
>
> [69] Chang et al. "Text to 3D Scene Generation with Rich Lexical Grounding." ACL 2015.
>
> [70] Chen et al. "Scene graph prediction with limited labels." ICCV 2019.
>
> [71] Qiuhong et al. "LEGO-Net: Learning Regular Rearrangements of Objects in Rooms". CVPR 2023.

---

> > ### Comment · Reviewer_Wi3K · 2023-08-16
> > **Rebuttal Acknowledgement**
> >
> > Thanks for the responses to my questions.
> >
> > I have a few questions still.
> >
> > 1) Complexity and Motivation -- First, a prior work from CVPR 2020 (called Total3D) and a follow-up work from ICCV 2021 (called Implicit3DUnderstanding), both perform scene reconstruction. Both of them target the Layout-to-Scene problem while also generating the constituent shapes. The latter, however, uses a graph representation for the scene. I would consider this work to be similar to the latter, but now in the generation paradigm. So, this is not the first work to generate shapes as a part of the layout generation/recon goal. Second, let me assume that incorporating contextual inter-object relations with the layout information helps generate better 3D shapes. This is a really interesting claim that needs a lot of investigation. The questions I would ask are: How can I be sure that this is due to contextual relations rather than the knowledge base of the pre-trained latent diffusion model employed? Under what scenarios of the proposed framework will the generated shapes look incompatible? I do not see an effort in educating the readers about these questions or an effort in investigating these really interesting questions. Answering these questions would make the paper more informative.
> >
> > Again, I am not convinced as to why combine Layout and Shape generation in one work if enough novelty exists in the Text-to-Shape part of the work. There is enough interest in the community to see frameworks that do a good job of shape generation from text inputs, and solving this sub-problem effectively should be a novel project on its own. If the idea is to make a generative framework, I argue that the paper's contribution is mixing up many ideas into one single paper, with no real *takeaway* message. I currently see the paper as presenting a complex approach to solving the problem of 3D scene generation from scene graphs. Talking about the input (scene graphs), I will move on to my next point, Dataset Annotation.
> >
> > 2) Dataset Annotation: Getting textual annotations on scene graphs is a laborious and expensive process. While I understand that this is necessary for training neural networks in general, I don't see this paradigm of using annotated scene graphs for learning a generative model of 3D scenes to be useful/practical compared to a paradigm where the generation process is conditioned on a collection of reconstructed scenes from scene images, which are abundantly available and require no annotations. In my opinion, this is a limitation of this work, one that relies on textually annotated scene graphs.
> >
> > 3) Limitations: Right, one should always discuss the limitations of the work in the main paper rather than the Supplementary. Now, I went through the limitations sections (Section 6) in the supplementary material. This section lacks a serious discussion of the proposed approach's limitations. What it currently describes is the fallacies of the dataset and the information about object attributes (textures) that were not included as a part of object encoding. This is highly superficial, to say the least. It will be valuable to the readers if the paper discusses aspects of the proposed approach that make narrowed assumptions, if any, about the weaknesses of the technique, and aspects of *technical* details that could be improved. Then, there should be some high-level thoughts on how these technical weaknesses could be addressed. While I am writing this, I cannot help but think of the motivation of the work, as well as the dataset constraints needed to get this done.
> >
> > With all the above, I am not persuaded to change my opinion of the paper.

---

> > > ### Author Response · Authors · 2023-08-17
> > > **Thanks for letting us know your further concerns**
> > >
> > > We conclude and answer your concerns as follows:
> > > ## 1. About scene graphs
> > > >*The latter, however, uses a graph representation for the scene.*
> > >
> > > >*In my opinion, this is a limitation of this work, one that relies on textually annotated scene graphs.*
> > >
> > > The scene graphs in our paper are semantic scene graphs [1,2,3], which model the semantic relationships between objects as written in the Preliminary (p.3, Section 3). This differentiates our work from Implicit3DUnderstanding, which only models the relative geometry in their graphs.
> > >
> > > >*I don't see this paradigm of using annotated scene graphs [...], which are abundantly available and require no annotations.*
> > >
> > > There are more benefits of using a scene graph over sentences, as mentioned in the caption of Figure 1 in [2]. Our work provides an alternative to using such a compact, symbolic, yet structured input as a condition to generate 3D scenes. We have rebutted that one can easily and explicitly get such a scene graph from other modalities like images exactly as the reviewer mentioned or even use a GUI to have such information. Once the scene graph is extracted, our framework can generate plausible scenes.
> > > ## 2. The claim of our method
> > > >*[...] a follow-up work from ICCV 2021 (called Implicit3DUnderstanding) [...] this is not the first work to generate shapes as a part of the layout generation/recon goal*
> > >
> > > We want to point out that we did not claim that this is the first work that generates shapes together with layout generation.
> > > The focus of this work is modeling a continuous latent manifold which allows us to sample (potentially) multiple plausible scenes that are semantically and contextually coherent given scene graph conditions. This differentiates us from the Implicit3DUnderstanding, which aims for exact reconstruction matched to the shape information conveyed by RGB input.
> > >
> > > >*let me assume that incorporating contextual inter-object relations with the layout information helps generate better 3D shapes. [...] How can I be sure that this is due to contextual relations rather than the knowledge base of the pre-trained latent diffusion model employed?*
> > >
> > > We also did not claim that the consideration of relations brings "better 3D shapes". We explicitly rebutted that it is the diffusion process bringing better qualities compared to prior work Graph-to-3D:
> > > *"Besides, the optimization from the diffusion-based shape branch not only assists layout generation during training but also brings better shape generation quality than previous work."*
> > >
> > > We evaluated the scene-level appearance with FID/KID and provided detailed experiments (See p.8. Table 1). Compared with Graph-to-3D, our model shows better results benefiting from the diffusion process, which is now also supported by object-level MMD/COV/1-NNA. With the same reliance on the diffusion process as Layout+txt2shape, ours still yields more coherent results by considering the inter-object relations.
> > >
> > > ## 3. Clarification of our goal and contributions
> > > >*I argue that the paper's contribution is mixing up many ideas into one single paper [...]*
> > >
> > > >*Getting textual annotations on scene graphs is a laborious and expensive process.*
> > >
> > > **The task aspect:** Good shape-generation methods from text inputs are different from what we want to achieve. As stated in the paper and answered in our rebuttal, our goal is to achieve semantically and contextually coherent scene generation.
> > >
> > > **The method aspect:** To achieve the goal, we enrich the original scene graph with contextual information and leverage diffusion models conditioned on the inter-object relations to generate scenes by joint training and optimizing the layout and shape branches. As also mentioned by Reviewer zttF, the joint training of VAE and diffusion together is non-trivial.
> > >
> > > **The data aspect:** The annotation is not part of the application limitation. The laborious annotation is exactly one of our contributions to the community.
> > >
> > > ## 4. Limitations and potential improvement
> > > >*It will be valuable to the readers [...] about the weaknesses of the technique, and aspects of technical details that could be improved.*
> > >
> > > We discussed that the interpenetrating phenomena in 3D-FRONT prohibit our framework from achieving fully collision-free generation. We have also provided a PDF to show the qualitative examples. However, a possible thought is to introduce an additional IoU loss to alleviate such problems with the support of training on most clean scenes. Second, the texture renderer can be leveraged from the related part in CC3D [4]. We will move related words to the main paper in the final version.
> > >
> > > [1] Chang et al. "A comprehensive survey of scene graphs: Generation and application." T-PAMI 2021.
> > >
> > > [2] Johnson et al. "Image generation from scene graphs." CVPR 2018.
> > >
> > > [3] Wu et al. "Incremental 3D Semantic Scene Graph Prediction from RGB Sequences." CVPR 2023.
> > >
> > > [4] Bahmani et al. "Cc3d: Layout-conditioned generation of compositional 3d scenes." ICCV 2023.

---

### Author Rebuttal · Authors · 2023-08-09

# Thank you for your insightful comments!
We would like to thank all reviewers for their insightful and valuable comments. In summary, they highlighted the significance of the work (*“scene generation is an important problem”* (xttF), *“structured input modality gives rise to many applications”* (wi3K), and *“this is a nice work”* (ANPx)), indicated that *“architecture is novel”* (ANPx), *“this paper addresses it with a responsible model using modern components in a competent manner”* (xttF), *“the generated scenes seem to produce plausible outputs”* (wi3K). They recognized the value of SG-FRONT *“enriching an existing dataset with scene graph annotations”* (NT6V), *“large amount of additional labels for 3D-FRONT”* (xttF), and acknowledged our code and dataset publicity (NT6V). Furthermore, they appreciated our supplementary video, *“particularly liked the intuitive explanation”* (ANPx).

This rebuttal addresses each reviewer’s concerns. **We also appreciate the pointers for the grammar errors and typos, which we have since corrected. We further attach a PDF document in this rebuttal for reference.**

---

### Comment · Area_Chair_kNmt · 2023-08-20
**Thanks for your response -- AC Comment**

Thanks for providing your responses to the reviewers' comments. They are comprehensive and answer many of the concerns raised during the initial review phase. The discussion with other reviewers also provides additional context for the questions raised during the initial review.

-- Your AC

---

### Decision · Program_Chairs · 2023-09-21

**Decision:**

Accept (poster)

**Comment:**

The work received primarily positive reviews, with many reviewers recommending acceptance. All reviewers acknowledged the contributions made by the work in alleviating the need for large libraries of 3D assets. Several important questions were raised during the review period, including ablations, limitations, and claims made in the paper. The rebuttal seems to have addressed many of these concerns effectively. The authors are highly encouraged to incorporate the results and discussions from the rebuttal into the final version for completeness.